# DeepKD: A Deeply Decoupled and Denoised Knowledge Distillation Trainer

**Haiduo Huang**[*], **Jiangcheng Song**[*], **Yadong Zhang**,[*] **Pengju Ren**[†]
Institute of Artificial Intelligence and Robotics, Xi'an Jiaotong University
{huanghd,enone,3488928835}@stu.xjtu.edu.cn, pengjuren@xjtu.edu.cn

## Abstract

Recent advances in knowledge distillation have emphasized the importance of decoupling different knowledge components. While existing methods utilize momentum mechanisms to separate task-oriented and distillation gradients, they overlook the inherent conflict between target-class and non-target-class knowledge flows. Furthermore, low-confidence dark knowledge in non-target classes introduces noisy signals that hinder effective knowledge transfer. To address these limitations, we propose DeepKD, a novel training framework that integrates dual-level decoupling with adaptive denoising. First, through theoretical analysis of gradient signal-to-noise ratio (GSNR) characteristics in task-oriented and non-task-oriented knowledge distillation, we design independent momentum updaters for each component to prevent mutual interference. We observe that the optimal momentum coefficients for task-oriented gradient (TOG), target-class gradient (TCG), and non-target-class gradient (NCG) should be positively related to their GSNR. Second, we introduce a dynamic top-k mask (DTM) mechanism that gradually increases K from a small initial value to incorporate more non-target classes as training progresses, following curriculum learning principles. The DTM jointly filters low-confidence logits from both teacher and student models, effectively purifying dark knowledge during early training. Extensive experiments on CIFAR-100, ImageNet, and MS-COCO demonstrate DeepKD's effectiveness.

## 1 Introduction

Knowledge distillation (KD) has emerged as a powerful paradigm for model compression since its introduction by Hinton et al. [1], finding widespread adoption across computer vision [2; 3; 4] and NLP [5] domains. By transferring dark knowledge from large teacher models to compact student networks, KD addresses the critical challenge of deploying high-performance models on resource-constrained devices - a fundamental requirement for emerging applications like autonomous driving [6] and embodied AI systems [7; 8].

Recent advances in KD methodologies have primarily focused on three directions: (1) Multi-teacher ensemble distillation [9; 10] to enhance information transfer, (2) Intermediate feature distillation [11] through sophisticated alignment mechanisms, and (3) Input-space augmentation [12; 13] or output-space manipulation through noise injection [14; 15] and regularization [16; 17]. However, these approaches lack systematic analysis of two fundamental questions: *Which components of knowledge transfer contribute to student performance?  and How should different knowledge components be optimally coordinated during optimization?* While previous works have made significant progress - DKD [18] decouples KD loss into target class knowledge distillation (TCKD) and non-target class knowledge distillation (NCKD) components through loss reparameterization, revealing NCKD's

---

[*]Equal Contributions
[†]Corresponding Author

39th Conference on Neural Information Processing Systems (NeurIPS 2025).

crucial role in dark knowledge transfer, and DOT [19] introduces gradient momentum decoupling between task and distillation losses - critical limitations persist. First, existing methods fail to address the joint optimization of decoupled losses and their corresponding gradient momenta. Second, the theoretical foundation for momentum allocation lacks rigorous justification, relying instead on empirical observations of loss landscape.

To address these limitations, we present DeepKD, as illustrated in Figure 4, a knowledge distillation framework with theoretically grounded features. Our investigation begins with a comprehensive analysis of loss components and their corresponding optimization parameters in knowledge distillation. Through rigorous stochastic optimization analysis [20], we find that optimal momentum coefficients for task-oriented gradient (TOG), target-class gradient (TCG), and non-target-class gradient (NCG) components should be positively related to their gradient signal-to-noise ratio (GSNR) [21] in stochastic gradient descent optimizer with momentum [22]. This enables deep decoupling of optimization dynamics across different knowledge types.

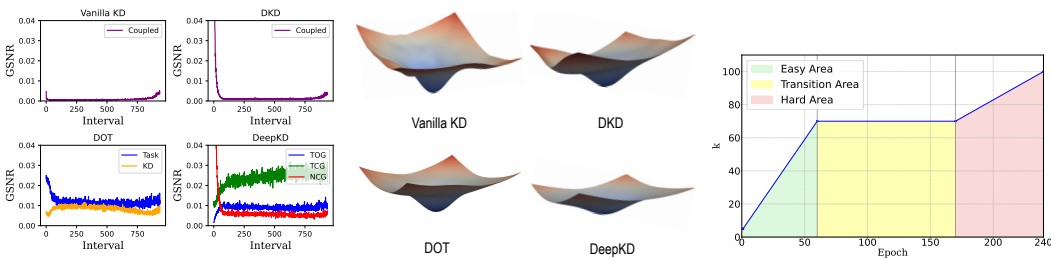

(a) GSNR during model training  (b) Loss landscape visualization  (c) Dynamic top-k masking process

Figure 1: Analysis of optimization dynamics and knowledge transfer of ResNet32×4/ResNet8×4 on CIFAR-100: (a) Gradient Signal-to-Noise Ratio (GSNR) comparison across different knowledge distillation methods, (b) Loss landscape visualization [23] showing the flatness of minima, and (c) Dynamic top-k masking process for dark knowledge denoising aligns with curriculum learning.

As shown in Figure 1(a), we visualize the GSNR of vanilla KD [1], DKD [18], DOT [19], and our proposed DeepKD throughout the training process (with gradient sampling at 200-iteration intervals). The results demonstrate that DeepKD further decouples KD gradients into TCG and NCG, achieving higher overall GSNR. This enhancement directly contributes to improved model generalization [21; 24]. Moreover, Figure 1(b) reveals that DeepKD exhibits a flatter loss landscape compared to other methods. This observation aligns with established findings that flatter minima in the loss landscape generally correlate with improved model generalization [25; 26]—a critical factor for effective knowledge distillation. Remarkably, through our GSNR-based deep gradient decoupling with momentum mechanisms alone, DeepKD achieves state-of-the-art performance across multiple benchmark datasets, as detailed in the experiments section.

Additionally, while prior works [1; 18; 19] emphasize the importance of teacher logits' dark knowledge, they typically process all non-target class logits uniformly. We challenge this convention through two key insights: (1) *Only non-target classes semantically adjacent to the target class provide meaningful and veritable dark knowledge.* (2) *Low-confidence logits may introduce optimization noise that outweighs their informational value.* To address these issues, we introduce a dynamic top-k mask (DTM) mechanism that progressively filters low-confidence logits (*i.e.*, potential noise sources) from both teacher and student outputs, implemented as a curriculum learning process [27], as shown in Figure 1(c). Unlike CTKD [28] which modulates task difficulty via a learnable temperature parameter, our method dynamically adjusts k from 5% of classes to full class count, balancing early-stage stability and late-stage refinement. Notably, while ReKD [29] applies top-k selection to target-similar classes but retains other non-target classes, we only preserve the top-k largest non-target-class logits based on the teacher and dynamically discard the remainder.

Comprehensive experiments across diverse model architectures and multiple benchmark datasets validate DeepKD's effectiveness. The framework demonstrates remarkable versatility by seamlessly integrating with existing logit-based distillation approaches, consistently achieving state-of-the-art performance across all evaluated scenarios.

## 2 Related Work

**Knowledge Distillation Paradigms:** Knowledge distillation has evolved along two main directions: feature-based and logit-based approaches. *Feature-based methods* transfer intermediate representations, starting with FitNets [30] using regression losses for hidden layer activations. This evolved through attention transfer [31] and relational distillation [32] to capture structural knowledge, culminating in multi-level alignment techniques like Chen *et al.*'s [33] multi-stage knowledge review and USKD's [34] normalized feature matching. While methods like FRS [35] and MDR [36] address teacher-student discrepancy through spherical normalization and adaptive stage selection, they often require complex feature transformations and overlook gradient-level interference. *Logit-based distillation*, pioneered by Hinton *et al.* [1], focuses on transferring dark knowledge through softened logits. Recent advances like DKD [18] decouple KD loss into target-class (TCKD) and non-target-class (NCKD) components, revealing NCKD's crucial role. Extensions including NTCE-KD [37] and MDR [36] enhance non-target class utilization but neglect gradient-level optimization dynamics.

**Theoretical Foundations and Methodological Advances:** Recent advances in knowledge distillation have explored both optimization strategies and theoretical foundations. On the optimization front, DOT [19] employs momentum mechanisms for gradient decoupling, CTKD [28] uses curriculum temperature scheduling, and ReKD [29] implements static top-k filtering, though these approaches often rely on empirical heuristics. Dark knowledge purification has been addressed through various strategies: TLLM [38] identifies undistillable classes via mutual information analysis, RLD [39] proposes logit standardization, and TALD-KD [14] combines target augmentation with logit distortion. The theoretical underpinnings of model generalization have been extensively studied through loss landscape geometry [25; 23], with Jelassi *et al.* [22] analyzing momentum's role in generalization.

Recent methodological advances have further enriched the knowledge distillation landscape. Niu *et al.* [40] propose respecting transfer gaps in knowledge distillation, while Huang *et al.* [41] introduce knowledge diffusion mechanisms for improved distillation. Li *et al.* [42] explore curriculum temperature scheduling for knowledge distillation, and Saidutta *et al.* [43] present controlled information flow approaches. Huang *et al.* [44] propose DIST+ with stronger adaptive teachers for enhanced knowledge transfer. Our work extends these principles by establishing the first theoretical connection between gradient signal-to-noise ratio (GSNR) and momentum allocation in KD, bridging optimization dynamics with knowledge transfer efficiency. Unlike DKD's loss-level decoupling or DOT's empirical momentum separation, we provide GSNR-driven theoretical guarantees for joint loss-gradient optimization. Compared to CTKD's temperature-centric curriculum or ReKD's static filtering, our dynamic top-k masking offers principled noise suppression while preserving semantic relevance, addressing the limitation of uniform processing of non-target logits in previous approaches.

## 3 Methodology

### 3.1 Preliminaries

**Vanilla KD:** Given a teacher model $\mathcal{T}$ and a student model $\mathcal{S}$, knowledge distillation transfers knowledge from $\mathcal{T}$ to $\mathcal{S}$ while maintaining performance. Let $\mathcal{X}$ be the input space and $\mathcal{Y}$ be the label space. For input $\mathbf{x} \in \mathcal{X}$, the models produce logits $\mathbf{z}^{\mathcal{T}} = \mathcal{T}(\mathbf{x})$ and $\mathbf{z}^{\mathcal{S}} = \mathcal{S}(\mathbf{x})$. The standard knowledge distillation [1] loss combines:

$$\mathcal{L}_{KD} = \alpha \mathcal{L}_{CE}(\sigma(\mathbf{z}^{\mathcal{S}}), \mathbf{y}) + (1 - \alpha)\tau^2 \mathcal{L}_{KL}(\sigma(\mathbf{z}^{\mathcal{S}}/\tau), \sigma(\mathbf{z}^{\mathcal{T}}/\tau)) \tag{1}$$

where $\sigma$ is the softmax function, $\mathcal{L}_{CE}$ is the cross-entropy loss with hard labels $\mathbf{y} \in \mathcal{Y}$, $\mathcal{L}_{KL}$ is the KL divergence between softened logits $\mathbf{p}^{\mathcal{T}}$ and $\mathbf{p}^{\mathcal{S}}$, *i.e.*, $\mathbf{p}^{\mathcal{T}} = \sigma(\mathbf{z}^{\mathcal{T}}/\tau)$ and $\mathbf{p}^{\mathcal{S}} = \sigma(\mathbf{z}^{\mathcal{S}}/\tau)$, $\tau$ is the temperature hyperparameter that controls the softness of the distribution, and $\alpha$ balances the losses.

**DKD:** Decoupled Knowledge Distillation (DKD) [18] splits the vanilla KD loss into target-class Knowledge Distillation (TCKD) and non-target-class Knowledge Distillation (NCKD) components:

$$\mathcal{L}_{DKD} = \alpha \mathcal{L}_{CE}(\mathbf{p}^{\mathcal{S}}, \mathbf{y}) + \tau^2(\beta_1 \mathcal{L}_{TCKD}(bp(p_t^{\mathcal{T}}), bp(p_t^{\mathcal{S}})) + \beta_2 \mathcal{L}_{NCKD}(\hat{\mathbf{p}}_{\backslash t}^{\mathcal{T}}, \hat{\mathbf{p}}_{\backslash t}^{\mathcal{S}})) \tag{2}$$

where $bp(.)$ is the *binary probabilities* function of the target class $p_t^{\mathcal{T}}(p_t^{\mathcal{S}})$, and all the other non-target classes $p_{\backslash t}^{\mathcal{T}}(p_{\backslash t}^{\mathcal{S}})$, and $\hat{\mathbf{p}}_{\backslash t}^{\mathcal{T}} = \sigma(\mathbf{z}_{\backslash t}^{\mathcal{T}}/\tau)$ and $\hat{\mathbf{p}}_{\backslash t}^{\mathcal{S}} = \sigma(\mathbf{z}_{\backslash t}^{\mathcal{S}}/\tau)$. $\mathcal{L}_{TCKD}$ transfers target class knowledge and $\mathcal{L}_{NCKD}$ captures relationships between non-target classes, revealing NCKD's importance in KD.

**DOT:** Distillation-Oriented Trainer (DOT) [19] maintains separate momentum buffers for cross-entropy and distillation loss gradients. For each mini-batch, DOT computes gradients $\boldsymbol{g}_{\text{ce}}$ and $\boldsymbol{g}_{\text{kd}}$ from

$\mathcal{L}_{\mathrm{CE}}$ and $\mathcal{L}_{\mathrm{KD}}$ respectively, then updates momentum buffers $\boldsymbol{v}_{\mathrm{ce}}$ and $\boldsymbol{v}_{\mathrm{kd}}$ with different coefficients:

$$\boldsymbol{v}_{\mathrm{ce}} \leftarrow \boldsymbol{g}_{\mathrm{ce}} + (\mu - \Delta)\boldsymbol{v}_{\mathrm{ce}}; \quad \boldsymbol{v}_{\mathrm{kd}} \leftarrow \boldsymbol{g}_{\mathrm{kd}} + (\mu + \Delta)\boldsymbol{v}_{\mathrm{kd}} \tag{3}$$

where $\mu$ is the base momentum and $\Delta$ is a hyperparameter controlling the momentum difference. By applying larger momentum to distillation loss gradients, DOT enhances knowledge transfer while mitigating optimization trade-offs between task and distillation objectives. However, DOT has two key limitations: (1) it fails to address the inherent conflict between target-class and non-target-class knowledge flows, which can lead to suboptimal optimization trajectories; and (2) it lacks a systematic analysis of the optimization dynamics across different loss components and their corresponding gradient momenta, particularly in handling low-confidence dark knowledge that introduces noisy signals and impedes effective knowledge transfer.

### 3.2 GSNR-Driven Momentum Allocation

Unlike previous DOT [19] that only decouples task and distillation gradients at a single level, our approach introduces a dual-level decoupling strategy further decomposing the gradient of the student model's training loss into three components: task-oriented gradient (TOG), target-class gradient (TCG), and non-target-class gradient (NCG). We define its gradient signal-to-noise ratio (GSNR) as:

$$\mathrm{GSNR} = \frac{\|\mathbb{E}[\nabla\mathcal{L}]\|_2^2}{\mathrm{Var}[\nabla\mathcal{L}]} = \frac{\|\mathbb{E}[\boldsymbol{g}]\|_2^2}{\mathbb{E}[\|\boldsymbol{g} - \mathbb{E}[\boldsymbol{g}]\|^2]} \approx \frac{\|\frac{1}{T}\sum_{t=1}^{T}\boldsymbol{g}_t\|_2^2}{\frac{1}{T}\sum_{t=1}^{T}\|\boldsymbol{g}_t - \frac{1}{T}\sum_{t=1}^{T}\boldsymbol{g}_t\|^2} \tag{4}$$

where $T$ is the sampling interval step size (default: 200), and $\boldsymbol{g}_t$ denotes the gradient at step $t$ including $\mathcal{TOG} = \nabla\mathcal{L}_{\mathrm{ce}}$, $\mathcal{TCG} = \nabla\mathcal{L}_{\mathrm{TCKD}}$, and $\mathcal{NCG} = \nabla\mathcal{L}_{\mathrm{NCKD}}$. For a target class $t$ and any class $i$, the gradients can be expressed as:

$$\mathcal{TOG}_i = \begin{cases} p_i^S - 1, & \text{if } i = t, \\ p_i^S, & \text{else} \end{cases}; \quad \mathcal{TCG}_i = \begin{cases} p_i^S - p_i^T, & \text{if } i = t, \\ -p_i^S \cdot (p_i^S - p_i^T), & \text{else} \end{cases}; \quad \mathcal{NCG}_i = \begin{cases} 0, & \text{if } i = t, \\ p_i^S - p_i^T, & \text{else} \end{cases} \tag{5}$$

where $p_i^S$ and $p_i^T$ denote the softmax probabilities of the student and teacher models respectively. The detailed mathematical derivation of this process is provided in Appendix A.2.

During stochastic optimization in deep neural networks, gradients computed at the end of each forward pass are used for backward propagation. These gradients form a sequence of stochastic vectors, where the statistical expectation and variance of the gradient can be estimated using the short-term sample mean within a temporal window [20]. Empirically, we find that gradient sampling at intervals of 200 iterations yields better performance. This estimation serves as the foundation for calculating the GSNR [45]. Specifically, the statistical expectation represents the true gradient direction, while the variance quantifies noise introduced by stochastic sampling.

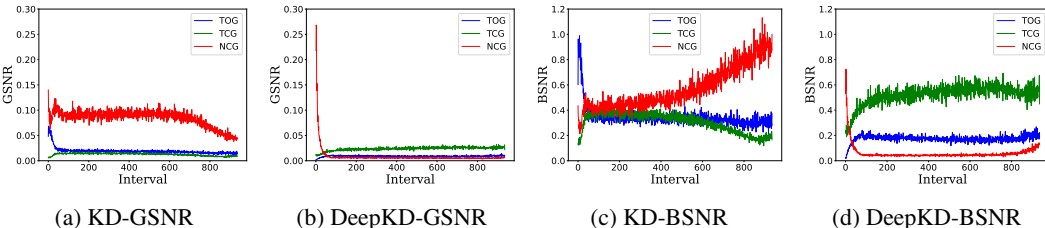

| (a) KD-GSNR | (b) DeepKD-GSNR | (c) KD-BSNR | (d) DeepKD-BSNR |

Figure 2: Comparison of gradient and buffer SNR between vanilla KD and DeepKD: (a) KD GSNR with less component separation, (b) DeepKD GSNR with better component distinction, (c) KD BSNR with limited separation, and (d) DeepKD BSNR with enhanced component differentiation.

To better understand the optimization dynamics, we conduct empirical analysis of GSNR curves of decoupled vanilla KD throughout the training process. As shown in Figure 2(a), vanilla KD exhibits consistently high signal-to-noise ratios (SNR) for non-target-class gradients (NCG) under identical momentum coefficients, indicating inherent difficulties in transferring "dark knowledge" to the student model. This persistent gradient divergence suggests unresolved conflicts between distillation objectives and target tasks, ultimately causing SNR instability in the gradient accumulation buffer (BSNR) (Figure 2(c)). Notably, gradient divergence reflects suboptimal convergence since well-converged models typically exhibit near-zero gradients, leading to smoothly decaying SNR

trajectories. We hypothesize that gradient components with higher SNR should be prioritized with heuristic weighting to help optimization. Figure 2(b) demonstrates that our DeepKD framework achieves accelerated and better absorption of dark knowledge from NCG while maintaining equilibrium among all gradient components. The resultant SNR trajectories exhibit smooth uniformity in the buffer (Figure 2(d)), validating our theoretical proposition. Empirical validation further corroborates that momentum coefficients for gradient components positively correlate with their respective SNRs during KD optimization. Crucially, experimental results reveal robustness to specific coefficient values, aligning with observations in prior work (DOT [19]).

Let's first examine the standard optimization formulation of SGD with momentum [46]:

$$\boldsymbol{v}_{t+1} = \boldsymbol{g}_t + \mu \boldsymbol{v}_t; \;\; \boldsymbol{\theta}_{t+1} = \boldsymbol{\theta}_t - \gamma \boldsymbol{v}_t \tag{6}$$

where $\boldsymbol{v}_t$ and $\boldsymbol{\theta}_t$ represent the momentum buffer and the model's trainable parameters at time step $t$, $\boldsymbol{g}_t$ is the current gradient, $\mu$ is the base momentum coefficient, and $\gamma$ is the learning rate. Through analysis of the GSNR in Figure 2(a), we observe that NCG and TOG maintain higher GSNR compared to TCG. This key observation motivates our adaptive momentum allocation strategy:

$$\boldsymbol{v}_{\text{TOG}} = \mathcal{TOG} + (\mu + \Delta)\boldsymbol{v}_{\text{TOG}}; \;\; \boldsymbol{v}_{\text{TCG}} = \mathcal{TCG} + (\mu - \Delta)\boldsymbol{v}_{\text{TCG}}; \;\; \boldsymbol{v}_{\text{NCG}} = \mathcal{NCG} + (\mu + \Delta)\boldsymbol{v}_{\text{NCG}} \tag{7}$$

where $\Delta$ is a hyperparameter controlling the momentum difference. As shown in Figure 2(b) & (d), our DeepKD with different momentum coefficients achieves significantly improved GSNR in both gradient buffers and raw gradients, further validating the necessity of our deep momentum decoupling approach for gradient components. Our GSNR-driven approach ensures each knowledge component follows its optimal optimization path while maintaining component independence, leading to more effective knowledge transfer. **Note that** our method is equally applicable to the Adam optimizer [47] by modifying only its first-order momentum, as validated on DeiT [48] (see Table 3).

### 3.3 Dynamic Top-K Masking

While existing advanced approaches [18; 19; 49] typically process non-target class logits through either uniform treatment or weighted separation [29], we identify two critical limitations in these conventional approaches: (1) Teacher models demonstrate extreme confidence in target class (softmax probabilities >0.99 for more than 92% of samples), while non-target classes collectively exhibit low confidence yet contain valuable dark knowledge, as evidenced in Figure 3(a). (2) The dark knowledge from non-target classes exhibits varying degrees of assimilability - classes semantically similar to the target (e.g., "tiger" for target "cat") provide beneficial dark knowledge, while semantically distant classes (e.g., "airplane") introduce noise and learning difficulties.

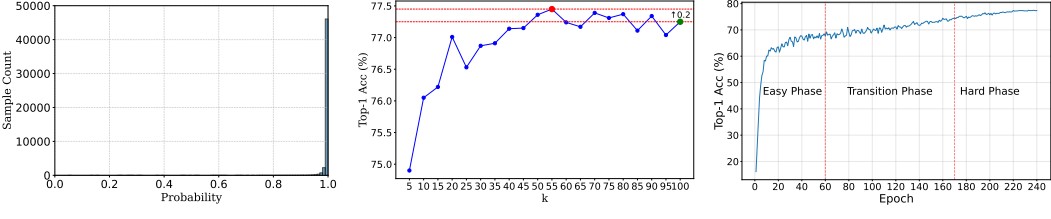

(a) Confidence of target class    (b) Accuracy of static top-k masking    (c) Training process of the best top-k

Figure 3: Analysis of top-k masking strategy. (a) Distribution of teacher model's confidence on target classes. (b) Accuracy comparison of different static top-k values for knowledge distillation. (c) Learning curve divided into distinct training phases with the optimal top-k masking approach.

To address these limitations, we first develop a **static top-k masking** approach that permanently filters classes with extreme semantic dissimilarity through fixed k-value masking, yielding baseline improvements as shown in Figure 3(b). Building upon this, we propose a more sophisticated **dynamic top-k masking** mechanism that implements phase-wise k-value scheduling inspired by curriculum learning [27]. This mechanism operates in three distinct phases via accuracy curves (Figure 3(c)):

- *Easy Learning Phase:* K increases linearly from 5% of the total number of classes to the optimal static K value
- *Transition Phase:* Maintains the optimal static K value
- *Hard Learning Phase:* Expands K linearly to encompass the full class count

The optimal static K value is determined through ablation studies or by using 20% of training data to reduce training cost. The complete process of dynamic top-k masking learning is illustrated in Figure 1(c). For each training iteration $i$, we compute the mask $\mathbf{M}_i$ as:

$$\mathbf{M}_i = \mathbb{I}(\mathrm{rank}(\mathbf{z}^{\mathcal{T}}_{\backslash t}) \leq K_i) \tag{8}$$

where $\mathrm{rank}(.)$ represents the rank of logits in ascending order, and $K_i$ gradually increases from 5% of classes to the total number of classes. The masked distillation loss is formulated as:

$$\mathcal{L}_{DTM} = \mathcal{L}_{NCKD}(\sigma(\mathbf{M}_i \odot \mathbf{z}^{\mathcal{S}}_{\backslash t}/\tau), \sigma(\mathbf{M}_i \odot \mathbf{z}^{\mathcal{T}}_{\backslash t}/\tau)) \tag{9}$$

where $\odot$ denotes element-wise multiplication. This mechanism effectively suppresses noise while preserving semantically relevant dark knowledge.

## 3.4 DeepKD Framework

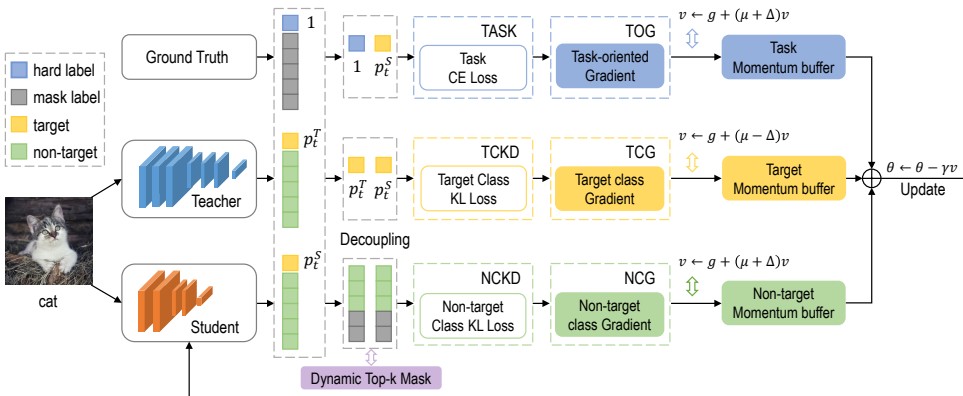

Figure 4: Detailed architecture of our DeepKD framework. Input images flow through teacher and student networks, producing target (yellow) and non-target (green) logits. The framework uses three independent gradient paths (task-oriented, target-class, and non-target-class) with separate momentum buffers. Dynamic Top-k Mask filters low-confidence non-target logits (gray cells).

Building upon theoretical analysis of GSNR, we propose DeepKD, a comprehensive framework that introduces deeply decoupled optimization with adaptive denoising for knowledge distillation. As illustrated in Figure 4, DeepKD decomposes the knowledge transfer process into three parallel gradient flows: task-oriented gradient (TOG), target-class gradient (TCG), and non-target-class gradient (NCG). Each gradient flow is managed independently with its own momentum buffer and optimized based on their distinct GSNR properties. This decoupled architecture enables more effective knowledge transfer by allowing each component to be optimized independently. The complete loss function of DeepKD (see Algorithm 1 in the Appendix for detailed implementation):

$$\mathcal{L}_{DeepKD} = \alpha \mathcal{L}_{CE}(\mathbf{p}^{\mathcal{S}}, \mathbf{y}) + \tau^2(\beta_1 \mathcal{L}_{TCKD}(bp(p^{\mathcal{T}}_t), bp(p^{\mathcal{S}}_t)) + \beta_2 \mathcal{L}_{DTM}) \tag{10}$$

where $\mathcal{L}_{CE}$ represents the standard cross-entropy loss, $\alpha$ and $\beta_i$ are fixed coefficients that balance the contribution of each loss component, $\mathcal{L}_{TCKD}$ is the target class loss, and $\mathcal{L}_{DTM}$ is the dynamic top-k masking loss of the non-target classes. This formulation enables the framework to effectively combine task-specific learning with knowledge distillation while maintaining computational efficiency.

# 4 Experiments

## 4.1 Datasets and Implementation

We conduct comprehensive evaluations on three widely-used benchmarks: CIFAR-100 [50] (100 classes, 50k training/10k validation $32 \times 32$ images), ImageNet-1K [51] (1,000 classes, 1.28M/50k images cropped to $224 \times 224$), and MS-COCO [52] (80-class detection, 118k training/5k validation images). For implementation, we follow standard practices using SGD optimizer with momentum 0.9 and weight decay of $5 \times 10^{-4}$ (CIFAR) or $1 \times 10^{-4}$ (ImageNet). The training schedule varies by dataset: CIFAR uses 240 epochs with batch size 64 and initial learning rate 0.01-0.05, while ImageNet uses 100 epochs with batch size 512 and learning rate 0.2. All experiments were conducted on a system equipped with an Nvidia RTX 4090 GPU and an AMD 64-Core Processor CPU.

Table 1: Top-1 Accuracy (%) on CIFAR-100 validation set. Results show homogeneous distillation (same architecture, different capacity) across feature-based and logit-based methods. Performance gains from our DeepKD framework are highlighted in blue and red.

| Type | | ResNet32×4
79.42 | VGG13
74.64 | WRN-40-2
75.61 | WRN-40-2
75.61 | ResNet56
72.34 | ResNet110
74.31 | ResNet110
74.31 |
|---|---|---|---|---|---|---|---|---|
| Type | Teacher | | | | | | | |
| | Student | ResNet8×4
72.50 | VGG8
70.36 | WRN-40-1
71.98 | WRN-16-2
73.26 | ResNet20
69.06 | ResNet32
71.14 | ResNet20
69.06 |
| Feature | FitNet [30] | 73.50 | 71.02 | 72.24 | 73.58 | 69.21 | 71.06 | 68.99 |
| | SimKD [53] | 78.08 | 74.89 | 74.53 | 75.53 | 71.05 | 73.92 | 71.06 |
| | CAT-KD [54] | 76.91 | 74.65 | 74.82 | 75.60 | 71.62 | 73.62 | 71.37 |
| Logit | KD [1] | 73.33 | 72.98 | 73.54 | 74.92 | 70.66 | 73.08 | 70.67 |
| | KD+DOT [19] | 74.98 | 73.77 | 73.87 | 75.43 | 71.11 | 73.37 | 70.97 |
| | KD+LSKD [49] | 76.62 | 74.36 | 74.37 | 76.11 | 71.43 | 74.17 | 71.48 |
| | KD+Ours (w/o top-k) | $76.69_{+3.36}$ | $74.96_{+1.98}$ | $74.80_{+1.26}$ | $76.14_{+1.22}$ | $71.79_{+1.13}$ | $74.20_{+1.12}$ | $71.59_{+0.92}$ |
| | KD+Ours (w. top-k) | $77.03_{+3.70}$ | $75.12_{+2.14}$ | $75.05_{+1.51}$ | $76.45_{+1.53}$ | $71.90_{+1.24}$ | $74.35_{+1.27}$ | $71.82_{+1.15}$ |
| | DKD [18] | 76.32 | 74.68 | 74.81 | 76.24 | 71.97 | 74.11 | 70.99 |
| | DKD+DOT [19] | 76.03 | 74.86 | 74.49 | 75.42 | 71.12 | 73.57 | 71.58 |
| | DKD+LSKD [49] | 77.01 | 74.81 | 74.89 | 76.39 | 72.32 | 74.29 | 71.48 |
| | DKD+Ours (w/o top-k) | $77.25_{+0.93}$ | $75.09_{+0.41}$ | $75.24_{+0.43}$ | $76.48_{+0.24}$ | $72.86_{+0.89}$ | $74.32_{+0.21}$ | $72.06_{+1.07}$ |
| | DKD+Ours (w. top-k) | $77.54_{+1.22}$ | $75.19_{+0.51}$ | $75.42_{+0.93}$ | $76.72_{+1.30}$ | $73.05_{+1.93}$ | $74.48_{+0.91}$ | $72.28_{+1.70}$ |
| | MLKD [55] | 77.08 | 75.18 | 75.35 | 76.63 | 72.19 | 74.11 | 71.89 |
| | MLKD+DOT [19] | 76.06 | 74.96 | 74.38 | 75.72 | 71.41 | 73.83 | 71.65 |
| | MLKD+LSKD [49] | 78.28 | 75.22 | 75.56 | 76.95 | 72.33 | 74.32 | 72.27 |
| | MLKD+Ours (w/o top-k) | $78.81_{+1.73}$ | $76.21_{+1.03}$ | $77.45_{+2.10}$ | $78.15_{+1.52}$ | $73.75_{+1.56}$ | $75.88_{+1.77}$ | $73.03_{+1.14}$ |
| | MLKD+Ours (w. top-k) | $79.15_{+2.07}$ | $76.45_{+1.27}$ | $77.82_{+2.47}$ | $\mathbf{78.49}_{+1.86}$ | $\mathbf{74.12}_{+1.93}$ | $76.15_{+2.04}$ | $73.28_{+1.39}$ |
| | CRLD [13] | 77.60 | 75.27 | 75.58 | 76.45 | 72.10 | 74.42 | 72.03 |
| | CRLD+DOT [19] | 76.54 | 74.34 | 74.75 | 75.57 | 71.11 | 73.91 | 70.67 |
| | CRLD+LSKD [49] | 78.23 | 74.74 | 76.28 | 76.92 | 72.09 | 75.16 | 72.26 |
| | CRLD+Ours (w/o top-k) | $78.90_{+1.30}$ | $76.29_{+1.02}$ | $76.98_{+1.40}$ | $77.99_{+1.54}$ | $73.29_{+1.19}$ | $76.03_{+1.61}$ | $73.07_{+1.04}$ |
| | CRLD+Ours (w. top-k) | $\mathbf{79.25}_{+1.65}$ | $\mathbf{76.58}_{+1.31}$ | $77.35_{+1.77}$ | $78.42_{+1.97}$ | $73.85_{+1.75}$ | $\mathbf{76.48}_{+2.06}$ | $\mathbf{73.52}_{+1.49}$ |

## 4.2 Image Classification

**Results on CIFAR-100.** Our DeepKD framework shows consistent improvements in both homogeneous and heterogeneous distillation settings. On homogeneous architectures (Table 1), DeepKD achieves accuracy gains of **+0.61%–+3.70%** without top-k masking and **+0.67%–+3.70%** with masking, outperforming feature-based methods by **1.2–4.8%**. The top-k variant further improves performance by up to **+1.86%**, with MLKD+Ours reaching **79.15%** accuracy. In heterogeneous scenarios (Table 2), DeepKD shows strong generalization: CRLD+Ours achieves **72.85%** for ResNet32×4→MobileNet-V2 (**+2.48%**), while KD+Ours attains **77.15%** for WRN-40-2→ResNet8×4 (**+3.18%**). Performance remains stable (*variance* ≤0.5%) under hyperparameter variations, confirming DeepKD's effectiveness in handling conflicting distillation signals.

**Results on ImageNet-1K.** As shown in Table 3, DeepKD achieves significant improvements across diverse teacher-student pairs. For ResNet50→MobileNet-V1, KD+Ours (w. top-k) boosts top-1 accuracy by **+4.15%** (74.65% vs. 70.50%), the largest gain among all configurations. CRLD+Ours (w. top-k) establishes new state-of-the-art results: **73.34%** (ResNet34→ResNet18, **+0.97%**) and **75.75%** (RegNetY-16GF→Deit-Tiny, **+1.89%**). The dynamic top-k masking consistently enhances performance, contributing additional gains of **+0.30%–+0.84%** in top-5 accuracy. Notably, our framework demonstrates strong scalability: (1) For lightweight students (MobileNet-V1/Deit-Tiny), improvements reach **+3.82%–+4.15%**; (2) With large teachers (RegNetY-16GF), top-5 accuracy exceeds **93.85%** (CRLD+Ours), surpassing all feature-based methods. These results validate DeepKD's effectiveness in large-scale distillation scenarios.

## 4.3 Object Detection on MS-COCO

DeepKD shows strong performance on object detection (see Table 4). With dynamic top-k masking (†), our method improves baseline KD by **+1.93% AP** (32.16% vs. 30.13%) and exceeds ReviewKD's 33.71% AP using only logit distillation. DKD+Ours† achieves **34.20% AP**, the best among all KD variants, with **+1.86%** gain over vanilla DKD. The dynamic top-k mechanism provides additional improvements of +0.05%-0.34% AP, with the largest boost in $AP_{75}$ (**+2.64%** for KD†). Our approach demonstrates better localization than feature-based LSKD, reaching **36.59% $AP_{75}$** (vs. LSKD's 36.34%) for DKD†. These results confirm DeepKD's effectiveness for dense prediction tasks.

Table 2: Top-1 Accuracy (%) on CIFAR-100 validation set with heterogeneous teacher-student pairs. Methods are grouped by type (feature/logit-based). Performance gains are shown in blue and red.

| Type | | ResNet32×4
79.42
SHN-V2
71.82 | ResNet32×4
79.42
WRN-16-2
73.26 | ResNet32×4
79.42
WRN-40-2
75.61 | WRN-40-2
75.61
ResNet8×4
72.50 | WRN-40-2
75.61
MN-V2
64.60 | VGG13
74.64
MN-V2
64.60 | ResNet50
79.34
MN-V2
64.60 |
|---|---|---|---|---|---|---|---|---|
| | **Teacher / Student** | | | | | | | |
| Feature | ReviewKD [56] | 77.78 | 76.11 | 78.96 | 74.34 | 71.28 | 70.37 | 69.89 |
| | SimKD [53] | 78.39 | 77.17 | 79.29 | 75.29 | 70.10 | 69.44 | 69.97 |
| | CAT-KD [54] | 78.41 | 76.97 | 78.59 | 75.38 | 70.24 | 69.13 | 71.36 |
| Logit | KD [1] | 74.45 | 74.90 | 77.70 | 73.97 | 68.36 | 67.37 | 67.35 |
| | KD+DOT [19] | 75.55 | 75.04 | 77.34 | 75.96 | 68.36 | 68.15 | 68.46 |
| | KD+LSKD [49] | 75.56 | 75.26 | 77.92 | 77.11 | 69.23 | 68.61 | 69.02 |
| | KD+Ours (w/o top-k) | $76.14_{+1.69}$ | $75.88_{+0.98}$ | $78.38_{+0.68}$ | $76.69_{+2.72}$ | $69.39_{+1.03}$ | $69.36_{+1.99}$ | $69.13_{+1.78}$ |
| | KD+Ours (w. top-k) | $76.45_{+2.00}$ | $76.12_{+1.22}$ | $78.65_{+0.95}$ | $77.15_{+3.18}$ | $69.85_{+1.49}$ | $69.92_{+2.55}$ | $69.78_{+2.43}$ |
| | DKD [18] | 77.07 | 75.70 | 78.46 | 75.56 | 69.28 | 69.71 | 70.35 |
| | DKD+DOT [19] | 77.41 | 75.69 | 78.42 | 75.71 | 62.32 | 68.89 | 70.12 |
| | DKD+LSKD [49] | 77.37 | 76.19 | 78.95 | 76.75 | 70.01 | 69.98 | 70.45 |
| | DKD+Ours (w/o top-k) | $77.68_{+0.61}$ | $76.61_{+0.91}$ | $79.57_{+1.11}$ | $76.86_{+1.30}$ | $70.29_{+1.01}$ | $70.04_{+0.33}$ | $70.48_{+0.13}$ |
| | DKD+Ours (w. top-k) | $77.95_{+0.88}$ | $76.89_{+1.19}$ | $79.82_{+1.36}$ | $76.90_{+1.34}$ | $70.65_{+1.37}$ | $70.38_{+0.67}$ | $70.72_{+0.37}$ |
| | MLKD [55] | 78.44 | 76.52 | 79.26 | 77.33 | 70.78 | 70.57 | 71.04 |
| | MLKD+DOT [19] | 78.53 | 75.82 | 79.01 | 76.53 | 69.15 | 68.26 | 67.73 |
| | MLKD+LSKD [49] | 78.76 | 77.53 | 79.66 | 77.68 | 71.61 | 70.94 | 71.19 |
| | MLKD+Ours (w/o top-k) | $80.55_{+2.11}$ | $78.28_{+1.76}$ | $81.40_{+2.14}$ | $78.31_{+0.98}$ | $72.17_{+1.39}$ | $72.46_{+1.89}$ | $73.04_{+2.00}$ |
| | MLKD+Ours (w. top-k) | $\mathbf{80.92}_{+2.48}$ | $78.65_{+2.13}$ | $81.78_{+2.52}$ | $78.49_{+1.16}$ | $72.53_{+1.75}$ | $\mathbf{72.82}_{+2.25}$ | $\mathbf{73.40}_{+2.36}$ |
| | CRLD [13] | 78.27 | 76.92 | 80.21 | 77.28 | 70.37 | 70.39 | 71.36 |
| | CRLD+DOT [19] | 78.33 | 75.97 | 79.41 | 76.41 | 64.36 | 61.35 | 69.96 |
| | CRLD+LSKD [49] | 78.61 | 77.37 | 80.58 | 78.03 | 71.52 | 70.48 | 71.43 |
| | CRLD+Ours (w/o top-k) | $79.72_{+1.45}$ | $78.79_{+1.87}$ | $81.82_{+1.61}$ | $78.62_{+1.34}$ | $72.09_{+1.72}$ | $71.99_{+1.60}$ | $72.01_{+0.65}$ |
| | CRLD+Ours (w. top-k) | $80.15_{+1.88}$ | $\mathbf{79.25}_{+2.33}$ | $\mathbf{82.35}_{+1.77}$ | $\mathbf{79.18}_{+1.90}$ | $\mathbf{72.85}_{+2.48}$ | $72.65_{+2.26}$ | $72.78_{+1.42}$ |

Table 3: Accuracy (%) on ImageNet-1K validation set. N/A indicates that the data is not available.

| Type | Teacher/Student | ResNet34/ResNet18 | | ResNet50/MN-V1 | | RegNetY-16GF/Deit-Tiny | |
|---|---|---|---|---|---|---|---|
| | Accuracy | top-1 | top-5 | top-1 | top-5 | top-1 | top-5 |
| | Teacher | 73.31 | 91.42 | 76.16 | 92.86 | 82.89 | 96.33 |
| | Student | 69.75 | 89.07 | 68.87 | 88.76 | 72.20 | 91.10 |
| Feature | SimKD [53] | 71.59 | 90.48 | 72.25 | 90.86 | N/A | N/A |
| | CAT-KD [54] | 71.26 | 90.45 | 72.24 | 91.13 | N/A | N/A |
| Logit | KD [1] | 71.03 | 90.05 | 70.50 | 89.80 | 73.15 | 91.85 |
| | KD+DOT [19] | 71.72 | 90.30 | 73.09 | 91.11 | 73.42 | 92.10 |
| | KD+LSKD [49] | 71.42 | 90.29 | 72.18 | 90.80 | 73.27 | 91.95 |
| | KD+Ours (w/o top-k) | $72.41_{+1.38}$ | $91.05_{+1.00}$ | $74.32_{+3.82}$ | $91.94_{+2.14}$ | $74.36_{+1.21}$ | $92.85_{+1.00}$ |
| | KD+Ours (w. top-k) | $72.85_{+1.82}$ | $91.35_{+1.30}$ | $74.65_{+4.15}$ | $92.25_{+2.45}$ | $74.83_{+1.68}$ | $93.15_{+1.30}$ |
| | DKD [18] | 71.70 | 90.41 | 72.05 | 91.05 | 73.35 | 92.05 |
| | DKD+DOT [19] | 72.03 | 90.50 | 73.33 | 91.22 | 73.66 | 92.25 |
| | DKD+LSKD [49] | 71.88 | 90.58 | 72.85 | 91.23 | 73.48 | 92.15 |
| | DKD+Ours (w/o top-k) | $72.78_{+1.08}$ | $90.96_{+0.55}$ | $74.41_{+2.36}$ | $92.08_{+1.03}$ | $74.57_{+1.22}$ | $93.07_{+1.02}$ |
| | DKD+Ours (w. top-k) | $73.15_{+1.45}$ | $91.25_{+0.84}$ | $74.43_{+2.38}$ | $91.95_{+0.90}$ | $74.95_{+1.60}$ | $93.36_{+1.31}$ |
| | MLKD [55] | 71.90 | 90.55 | 73.01 | 91.42 | 73.54 | 92.25 |
| | MLKD+DOT [19] | 70.94 | 90.15 | 71.65 | 90.28 | 73.25 | 91.95 |
| | MLKD+LSKD [49] | 72.08 | 90.74 | 73.22 | 91.59 | 73.78 | 92.45 |
| | MLKD+Ours (w/o top-k) | $73.18_{+1.28}$ | $91.23_{+0.68}$ | $74.77_{+1.76}$ | $92.35_{+0.93}$ | $75.15_{+1.61}$ | $93.48_{+1.03}$ |
| | MLKD+Ours (w. top-k) | $73.31_{+1.41}$ | $91.39_{+0.84}$ | $74.85_{+1.84}$ | $92.45_{+1.03}$ | $75.46_{+1.92}$ | $93.73_{+1.28}$ |
| | CRLD [13] | 72.37 | 90.76 | 73.53 | 91.43 | 73.82 | 92.55 |
| | CRLD+DOT [19] | 71.76 | 90.00 | 72.38 | 90.37 | 73.37 | 92.05 |
| | CRLD+LSKD [49] | 72.39 | 90.87 | 73.74 | 91.61 | 73.95 | 92.65 |
| | CRLD+Ours (w/o top-k) | $73.18_{+0.81}$ | $91.23_{+0.47}$ | $74.10_{+0.57}$ | $91.49_{+0.06}$ | $75.35_{+1.53}$ | $93.35_{+0.90}$ |
| | CRLD+Ours (w. top-k) | $\mathbf{73.34}_{+0.97}$ | $\mathbf{91.38}_{+0.62}$ | $\mathbf{74.85}_{+1.12}$ | $\mathbf{92.45}_{+1.02}$ | $\mathbf{75.75}_{+1.89}$ | $\mathbf{93.85}_{+1.40}$ |

## 5  Ablation Study

**Momentum Coefficients and Loss Functions.** Based on gradient signal-to-noise ratio (GSNR) analysis, we decompose the gradient momentum in DeepKD into Target-Class Gradient (TCG) and Non-Target-Class Gradient (NCG). To validate this decoupling strategy, we conduct comprehensive ablation studies on CIFAR-100 using ResNet32×4(teacher) and ResNet8×4(student) pairs (see Table 5). The results demonstrate the effectiveness of our approach and highlight the importance of each component. Notably, DeepKD introduces **only one hyper-parameter** $\Delta$, which proves

Table 4: Results on MS-COCO(val2017) based on Faster-RCNN-FPN[57]. Teacher-student pair is ResNet50 & MobileNet-V2. The values of columns with † based on dynamic top-k masking.

| Metric | Teacher | Student | ReviewKD | KD | KD+LSKD | KD+Ours | KD+Ours† | DKD | DKD+LSKD | DKD+Ours | DKD+Ours† |
|---|---|---|---|---|---|---|---|---|---|---|---|
| AP | 42.04 | 29.47 | 33.71 | 30.13 | 31.71 | $32.01_{+1.88}$ | $32.16_{+1.93}$ | 32.34 | 33.98 | $33.99_{+1.65}$ | $34.20_{+1.86}$ |
| $AP_{50}$ | 61.02 | 48.87 | 53.15 | 50.28 | 52.77 | $52.88_{+2.60}$ | $52.98_{+2.63}$ | 53.77 | 54.93 | $55.11_{+1.34}$ | $55.34_{+1.57}$ |
| $AP_{75}$ | 43.81 | 30.90 | 36.13 | 31.35 | 33.40 | $33.65_{+2.30}$ | $33.99_{+2.64}$ | 34.01 | 36.34 | $36.35_{+2.34}$ | $36.59_{+2.58}$ |

to be robust across different datasets. Following DOT [19], we set $\Delta$=0.075 for KD+DeepKD on CIFAR-100, and $\Delta$=0.05 for DKD+DeepKD, MLKD+DeepKD, and CRLD+DeepKD. For ImageNet, where teacher knowledge is more reliable, we use $\Delta$=0.05 for all variants. Our ablation studies on loss functions reveal that each component contributes significantly to the overall performance, with NCKD being the dominant factor—consistent with findings in DKD [18]. These results validate our decoupling strategy and demonstrate its effectiveness in improving model performance.

Table 5: Results of using our DeepKD+KD for Resnet32×4(teacher)/Resnet8×4(student) on CIFAR-100. Left: impact of momentum coefficients ($\Delta$); Middle: effectiveness of different loss combinations; Right: performance with dynamic top-k masking and curriculum learning.

| Momentum Coefficients | | | | | Loss Functions | | | | | Dynamic top-k with curriculum learning | | | | |
|---|---|---|---|---|---|---|---|---|---|---|---|---|---|---|
| $\Delta_{TOG}$ | $\Delta_{TCG}$ | $\Delta_{NCG}$ | top-1 | top-5 | TASK | TCKD | NCKD | top-1 | top-5 | k-value | Phase1 | Phase2 | top-1 | top-5 |
| 0.00 | 0.00 | 0.00 | 74.13 | 92.82 | ✗ | ✗ | ✗ | 1.21 | 5.31 | 55 | 40 | 170 | 76.98 | 91.62 |
| 0.075 | 0.00 | 0.00 | 74.89 | 93.27 | ✓ | ✗ | ✗ | 73.28 | 92.89 | 55 | 60 | 170 | 77.03 | 92.07 |
| 0.00 | 0.075 | 0.00 | 74.58 | 93.52 | ✗ | ✓ | ✗ | 74.58 | 93.52 | 55 | 60 | 160 | **77.31** | **93.38** |
| 0.00 | 0.00 | 0.075 | 75.25 | 93.61 | ✗ | ✗ | ✓ | 75.25 | 93.61 | 55 | 60 | 180 | 77.13 | 92.28 |
| 0.075 | 0.075 | 0.00 | 75.41 | 93.71 | ✓ | ✓ | ✗ | 71.50 | 93.11 | 60 | 40 | 170 | 77.20 | 92.51 |
| 0.00 | 0.075 | 0.075 | 76.21 | 93.90 | ✗ | ✓ | ✓ | 75.42 | 93.83 | 60 | 60 | 170 | 77.19 | 92.36 |
| 0.075 | 0.00 | 0.075 | 76.19 | 93.97 | ✓ | ✗ | ✓ | 76.06 | 94.12 | 60 | 60 | 160 | 77.29 | 93.12 |
| 0.075 | 0.075 | 0.075 | **76.69** | **94.21** | ✓ | ✓ | ✓ | **76.69** | **94.21** | 60 | 60 | 180 | 77.21 | 93.32 |

**Dynamic Top-k Masking.** To discard the impact of dynamic top-k masking, we conduct the above ablation studies on momentum coefficients and loss functions without this strategy. For the dynamic top-k masking configuration, we empirically set the parameters as k-value = 55, Phase1 = 60, and Phase2 = 170 (see Table 5). The experimental results demonstrate that even simple hyperparameter tuning can further improve model performance, suggesting that integrating a dynamic top-k masking mechanism into KD holds great potential and warrants further exploration in future work.

Table 6: Results of using our DeepKD with different distillation methods on CIFAR-100. We evaluate the impact of decoupled gradients (Decoupled) and dynamic top-k masking (DTM) strategies.

| Method | KD [1] | | | | DKD [18] | | | | MLKD [55] | | | | CRLD [13] | | | |
|---|---|---|---|---|---|---|---|---|---|---|---|---|---|---|---|---|
| Decoupled | ✗ | ✓ | ✗ | ✓ | ✗ | ✓ | ✗ | ✓ | ✗ | ✓ | ✗ | ✓ | ✗ | ✓ | ✗ | ✓ |
| DTM | ✗ | ✗ | ✓ | ✓ | ✗ | ✗ | ✓ | ✓ | ✗ | ✗ | ✓ | ✓ | ✗ | ✗ | ✓ | ✓ |
| top-1 | 73.33 | 76.69 | 74.89 | **77.03** | 76.32 | 77.25 | 76.58 | **77.54** | 77.08 | 78.81 | 77.41 | **79.15** | 77.60 | 78.90 | 77.93 | **79.25** |

Furthermore, we conduct ablation studies on both gradient decoupling and dynamic top-k masking strategies. The results demonstrate that each component individually enhances performance compared to the original distillation methods, while their combination yields further improvements. Note that when DTM is enabled individually, the mechanism operates on all logits including the target class.

# 6 Discussion

## 6.1 Computational Complexity Analysis

To address concerns about computational overhead, we provide a comprehensive analysis of DeepKD's training efficiency and resource requirements. Table 7 shows the memory consumption and training time overhead for different teacher-student pairs on CIFAR-100 and ImageNet-1K. DeepKD introduces minimal memory overhead, with increases of less than 1% on CIFAR-100 and less than 0.2% on ImageNet-1K. This modest increase primarily stems from storing student gradients during distillation, as the teacher model remains frozen. The efficient CUDA memory allocator further mitigates additional memory footprint, confirming DeepKD's suitability for large-scale training scenarios.

Table 7: Computational overhead analysis of DeepKD: All experiments use a single 2080Ti GPU for CIFAR-100 and two RTX 4090 GPUs for training on ImageNet-1K.

| Dataset | Method | Memory (MB) | | Training Time (Hours) | |
| --- | --- | --- | --- | --- | --- |
| | | Baseline | DeepKD | Baseline | DeepKD |
| CIFAR-100 | KD | 799 | 805 (+0.75%) | 1.6 | 2.6 (+62%) |
| | DKD | 799 | 805 (+0.75%) | 1.7 | 2.7 (+60%) |
| | MLKD | 983 | 987 (+0.41%) | 9.2 | 12.3 (+33%) |
| | CRLD | 981 | 985 (+0.41%) | 2.9 | 4.5 (+52%) |
| ImageNet-1K | KD | 21344 | 21370 (+0.12%) | 20.0 | 29.4 (+47%) |
| | DKD | 21350 | 21370 (+0.12%) | 20.1 | 29.8 (+48%) |
| | MLKD | 34910 | 34960 (+0.14%) | 34.2 | 53.1 (+55%) |
| | CRLD | 34862 | 34909 (+0.13%) | 32.2 | 51.2 (+59%) |

## 6.2 Training Time and Convergence Analysis

Table 8: Comparing training epochs and final accuracy. All baseline methods are trained for the standard number of epochs (240 for CIFAR-100, 480 for MLKD, 100 for ImageNet-1K).

| Dataset | Method | CIFAR-100 | | ImageNet-1K | |
| --- | --- | --- | --- | --- | --- |
| | | Epochs | Top-1 (%) | Epochs | Top-1 (%) |
| Baseline | KD | 240 | 73.33 | 100 | 71.03 |
| | DKD | 240 | 76.32 | 100 | 71.70 |
| | MLKD | 480 | 77.08 | 100 | 71.90 |
| | CRLD | 240 | 77.60 | 100 | 72.37 |
| DeepKD | KD+Ours | **160** | **76.83** | **65** | **71.76** |
| | DKD+Ours | **160** | **77.31** | **65** | **72.16** |
| | MLKD+Ours | **320** | **79.04** | **65** | **72.52** |
| | CRLD+Ours | **160** | **78.87** | **65** | **72.98** |

While DeepKD introduces moderate per-epoch training time increases, it achieves superior accuracy with substantially fewer epochs than baseline methods, resulting in a favorable overall time-accuracy trade-off. Table 8 demonstrates that DeepKD enables faster convergence while maintaining higher final performance. The additional training cost is justified by substantial performance gains. Since knowledge distillation is typically a one-time training process, investing more resources to achieve superior models is standard practice in the field. Importantly, this additional training cost does not affect the final inference speed of the student model, making DeepKD a highly efficient route to better-performing models.

## 7 Limitations and Future Work

While our current work focuses on logit-based distillation, the SNR-driven momentum decoupling mechanism naturally extends to feature distillation scenarios. By treating feature alignment losses as additional optimization components, our framework can automatically handle multi-level knowledge transfer without manual weighting, complementing existing feature enhancement techniques like attention transfer [31] and contrastive distillation [58]. Future work could explore the application of our framework to more complex scenarios, such as multi-teacher distillation and cross-modal knowledge transfer. Additionally, our dynamic top-k masking strategy shows promising results in improving distillation performance, suggesting potential for further refinement and adaptation to different model architectures and datasets.

## 8 Conclusion

This paper presents DeepKD, a novel knowledge distillation framework that introduces SNR-driven momentum decoupling to address the gradient conflict between task learning and knowledge transfer. Our approach automatically allocates appropriate momentum coefficients based on gradient SNR characteristics, enabling effective optimization of both task-specific and knowledge distillation objectives. Through extensive experiments on multiple datasets and model architectures, we demonstrate that DeepKD consistently improves the performance of various SOTA distillation methods.

# 9 Acknowledgments

This work was supported in part by National Natural Science Foundation of China No. 62088102 and No.62302381. The Authors are with the National Key Laboratory of Human-Machine Hybrid Augmented Intelligence, National Engineering Research Center of Visual Information and Applications, and Institute of Artificial Intelligence and Robotics, Xi'an Jiaotong University, Xi'an, Shaanxi, China.

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

# A  Technical Appendices and Supplementary Material

## A.1  Distillation fidelity and feature visualization.

To provide an intuitive understanding of our method's effectiveness, we visualize both the distillation fidelity and deep feature representations. Following DKD [18], we calculate the absolute distance between correlation matrices of the teacher (ResNet32×4) and student (ResNet8×4) on CIFAR-100. As shown in Figure 5, DeepKD enables the student to produce logits more similar to the teacher compared to other methods. Additionally, our feature visualizations in Figure 6 demonstrate that our pre-process enhances feature separability and discriminability across various distillation methods including KD, DKD, MLKD, and CRLD.

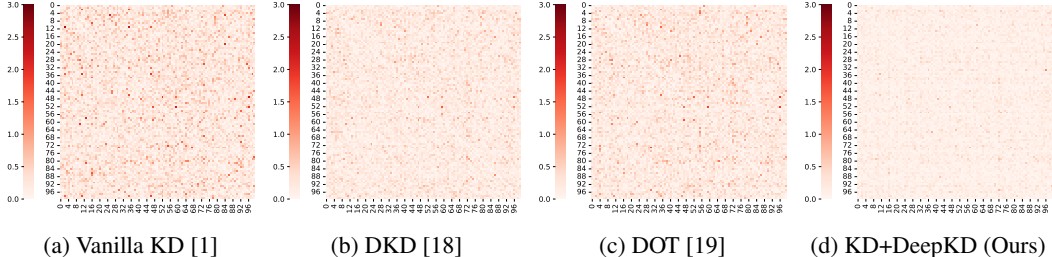

|  (a) Vanilla KD [1]  |  (b) DKD [18]  |  (c) DOT [19]  |  (d) KD+DeepKD (Ours)  |

Figure 5: Difference of student and teacher logits. DeepKD leads to a significantly smaller difference (more similar prediction) than other KD methods.

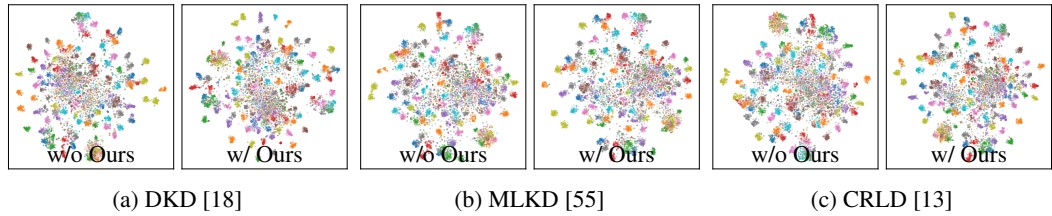

|  (a) DKD [18]  |  (b) MLKD [55]  |  (c) CRLD [13]  |

Figure 6: The t-SNE [59] feature visualization of ResNet32×4 and ResNet8×4 on CIFAR-100.

## A.2  Theoretical Analysis

Let us first define the key components of our knowledge distillation framework. Given a teacher model $\mathcal{T}$ and a student model $\mathcal{S}$, we aim to transfer knowledge from $\mathcal{T}$ to $\mathcal{S}$ while maintaining task performance. The overall loss function combines task-specific loss and knowledge distillation loss:

$$\mathcal{L} = \alpha\mathcal{L}_{CE}(\boldsymbol{p}^{\mathcal{S}}, \boldsymbol{p}^{\mathcal{G}}) + (1 - \alpha)\mathcal{L}_{KD}(\boldsymbol{p}^{\mathcal{S}}, \boldsymbol{p}^{\mathcal{T}}) \tag{11}$$

where $\alpha$ is a balancing parameter, $\boldsymbol{p}^{\mathcal{S}}$ and $\boldsymbol{p}^{\mathcal{T}}$ are the output probabilities of student and teacher models respectively, and $\boldsymbol{p}^{\mathcal{G}}$ represents the ground truth labels. The knowledge distillation loss $\mathcal{L}_{KD}$ can be elegantly decomposed into two components using KL divergence:

$$\mathcal{L}_{KD}(\boldsymbol{p}^{\mathcal{S}}, \boldsymbol{p}^{\mathcal{T}}) = KL(\boldsymbol{p}^{\mathcal{T}}||\boldsymbol{p}^{\mathcal{S}}) = KL(\boldsymbol{b}^{\mathcal{T}}||\boldsymbol{b}^{\mathcal{S}}) + (1 - p_t^{\mathcal{T}})KL(\hat{\boldsymbol{p}}^{\mathcal{T}}||\hat{\boldsymbol{p}}^{\mathcal{S}}) \tag{12}$$

Our DeepKD introduces a dual-level decoupling strategy further decomposes the gradient of the student model's training loss into three components: task-oriented gradient ($\mathcal{TOG}$), target-class gradient ($\mathcal{TCG}$), and non-target-class gradient ($\mathcal{NCG}$).

The probability computation:

$$p_i = \frac{e^{z_i}}{\sum\limits_{k} e^{z_k}} \tag{13}$$

where $z_i$ is the logit of the $i$-th class. And its derivatives are

$$
\begin{aligned}
\frac{\partial p_i}{\partial z_i} &= \frac{e^{z_i} \cdot \sum\limits_k e^{z_k} - e^{z_i} \cdot e^{z_i}}{(\sum\limits_k e^{z_k})^2} \\
&= \frac{e^{z_i}}{\sum\limits_k e^{z_k}} \cdot \frac{\sum\limits_k e^{z_k} - e^{z_i}}{\sum\limits_k e^{z_k}} \\
&= p_i \cdot (1 - p_i)
\end{aligned}
\tag{14}
$$

$$
\begin{aligned}
\frac{\partial p_i}{\partial z_j} &= \frac{0 \cdot \sum\limits_k e^{z_k} - e^{z_i} \cdot e^{z_j}}{(\sum\limits_k e^{z_k})^2} \\
&= -\frac{e^{z_i}}{\sum\limits_k e^{z_k}} \cdot \frac{e^{z_j}}{\sum\limits_k e^{z_k}} \\
&= -p_i \cdot p_j, \forall j \neq i
\end{aligned}
\tag{15}
$$

The task loss:

$$
\mathcal{L}_{task} = CE(\boldsymbol{p}^{\mathcal{G}}, \boldsymbol{p}^{\mathcal{S}}) = -log(p_t^{\mathcal{S}})
\tag{16}
$$

and its gradients are

$$
\begin{aligned}
\mathcal{TOG}_t &= \frac{\partial \mathcal{L}_{task}}{\partial z_t^{\mathcal{S}}} \\
&= \frac{\partial \mathcal{L}_{task}}{\partial p_t^{\mathcal{S}}} \cdot \frac{\partial p_t^{\mathcal{S}}}{\partial z_t^{\mathcal{S}}} \\
&= -\frac{1}{p_t^{\mathcal{S}}} \cdot (p_t^{\mathcal{S}} \cdot (1 - p_t^{\mathcal{S}})) \\
&= p_t^{\mathcal{S}} - 1
\end{aligned}
\tag{17}
$$

$$
\begin{aligned}
\mathcal{TOG}_j &= \frac{\partial \mathcal{L}_{task}}{\partial z_j^{\mathcal{S}}} \\
&= \frac{\partial \mathcal{L}_{task}}{\partial p_t^{\mathcal{S}}} \cdot \frac{\partial p_t^{\mathcal{S}}}{\partial z_j^{\mathcal{S}}} \\
&= -\frac{1}{p_t^{\mathcal{S}}} \cdot (-p_t^{\mathcal{S}} \cdot p_j^{\mathcal{S}}) \\
&= p_j^{\mathcal{S}}, \forall j \neq t
\end{aligned}
\tag{18}
$$

The binary probability is constructed as

$$
\boldsymbol{b} = [p_t, 1 - p_t]^T
\tag{19}
$$

The TCKD Loss:

$$
\begin{aligned}
KL(\boldsymbol{b}^{\mathcal{T}} || \boldsymbol{b}^{\mathcal{S}}) &= p_t^{\mathcal{T}} \cdot log\frac{p_t^{\mathcal{T}}}{p_t^{\mathcal{S}}} + (1 - p_t^{\mathcal{T}}) \cdot log\frac{1 - p_t^{\mathcal{T}}}{1 - p_t^{\mathcal{S}}} \\
&= -p_t^{\mathcal{T}} \cdot logp_t^{\mathcal{S}} - (1 - p_t^{\mathcal{T}}) \cdot log(1 - p_t^{\mathcal{S}}) + p_t^{\mathcal{T}} \cdot logp_t^{\mathcal{T}} + (1 - p_t^{\mathcal{T}}) \cdot log(1 - p_t^{\mathcal{T}}) \\
&= CE(\boldsymbol{b}^{\mathcal{T}}, \boldsymbol{b}^{\mathcal{S}}) - H(\boldsymbol{b}^{\mathcal{T}})
\end{aligned}
\tag{20}
$$

and its derivatives are

$$
\begin{aligned}
\mathcal{TCG}_t &= \frac{\partial KL(\boldsymbol{b}^{\mathcal{T}}||\boldsymbol{b}^{\mathcal{S}})}{\partial z_t^{\mathcal{S}}} \\
&= \frac{\partial KL(\boldsymbol{b}^{\mathcal{T}}||\boldsymbol{b}^{\mathcal{S}})}{\partial p_t^{\mathcal{S}}} \cdot \frac{\partial p_t^{\mathcal{S}}}{\partial z_t^{\mathcal{S}}} \\
&= (-\frac{p_t^{\mathcal{T}}}{p_t^{\mathcal{S}}} + \frac{1 - p_t^{\mathcal{T}}}{1 - p_t^{\mathcal{S}}}) \cdot (p_t^{\mathcal{S}} \cdot (1 - p_t^{\mathcal{S}})) \\
&= -p_t^{\mathcal{T}} \cdot (1 - p_t^{\mathcal{S}}) + (1 - p_t^{\mathcal{T}}) \cdot p_t^{\mathcal{S}} \\
&= p_t^{\mathcal{S}} - p_t^{\mathcal{T}}
\end{aligned}
\tag{21}
$$

$$
\begin{aligned}
\mathcal{TCG}_j &= \frac{\partial KL(\boldsymbol{b}^{\mathcal{T}}||\boldsymbol{b}^{\mathcal{S}})}{\partial z_j^{\mathcal{S}}} \\
&= \frac{\partial KL(\boldsymbol{b}^{\mathcal{T}}||\boldsymbol{b}^{\mathcal{S}})}{\partial p_t^{\mathcal{S}}} \cdot \frac{\partial p_t^{\mathcal{S}}}{\partial z_j^{\mathcal{S}}} \\
&= (-\frac{p_t^{\mathcal{T}}}{p_t^{\mathcal{S}}} + \frac{1 - p_t^{\mathcal{T}}}{1 - p_t^{\mathcal{S}}}) \cdot (-p_t^{\mathcal{S}} \cdot p_j^{\mathcal{S}}) \\
&= \frac{(p_t^{\mathcal{T}} \cdot p_j^{\mathcal{S}}) \cdot (1 - p_t^{\mathcal{S}}) + (1 - p_t^{\mathcal{T}}) \cdot (-p_t^{\mathcal{S}} \cdot p_j^{\mathcal{S}})}{1 - p_t^{\mathcal{S}}} \\
&= \frac{p_t^{\mathcal{T}} \cdot p_j^{\mathcal{S}} - p_t^{\mathcal{S}} \cdot p_j^{\mathcal{S}}}{1 - p_t^{\mathcal{S}}} \\
&= \frac{p_j^{\mathcal{S}}}{1 - p_t^{\mathcal{S}}} \cdot (p_t^{\mathcal{T}} - p_t^{\mathcal{S}}) \\
&= -\hat{p}_j^{\mathcal{S}} \cdot (p_t^{\mathcal{S}} - p_t^{\mathcal{T}}), \forall j \neq t
\end{aligned}
\tag{22}
$$

The non-target class probability distribution is calculated as:

$$
\boldsymbol{p}_{\setminus t} = [p_1, p_2, ..., p_{t-1}, p_{t+1}, ..., p_N]^T
\tag{23}
$$

and

$$
\begin{aligned}
\hat{\boldsymbol{p}} &= Normalize(\boldsymbol{p}_{\setminus t}) \\
&= \frac{\boldsymbol{P}_{\setminus t}}{1 - p_t}
\end{aligned}
\tag{24}
$$

Specifically, its components are

$$
\begin{aligned}
\hat{p}_i &= \frac{p_i}{1 - p_t} \\
&= \frac{p_i}{1 - p_t} \cdot \frac{\sum\limits_k e^{z_k}}{\sum\limits_k e^{z_k}} \\
&= \frac{e^{z_i}}{\sum\limits_{k \neq t} e^{z_k}}, \forall i \neq t
\end{aligned}
\tag{25}
$$

and its derivatives are

$$
\frac{\partial \hat{p}_i}{\partial z_t} = 0, \forall i \neq t
\tag{26}
$$

$$
\frac{\partial \hat{p}_i}{\partial z_i} = \hat{p}_i \cdot (1 - \hat{p}_i), \forall i \neq t
\tag{27}
$$

$$
\frac{\partial \hat{p}_i}{\partial z_j} = -\hat{p}_i \cdot \hat{p}_j, \forall i \neq t, j \neq t, j \neq i
\tag{28}
$$

The non-target class loss:

$$KL(\hat{\boldsymbol{p}}^{\mathcal{T}}||\hat{\boldsymbol{p}}^{\mathcal{S}}) = \sum_{\substack{i=1 \\ i \neq t}}^{N} \hat{p}_i^{\mathcal{T}} \cdot log\frac{\hat{p}_i^{\mathcal{T}}}{\hat{p}_i^{\mathcal{S}}}$$

$$= -\sum_{\substack{i=1 \\ i \neq t}}^{N} \hat{p}_i^{\mathcal{T}} \cdot log\hat{p}_i^{\mathcal{S}} + \sum_{\substack{i=1 \\ i \neq t}}^{N} \hat{p}_i^{\mathcal{T}} \cdot log\hat{p}_i^{\mathcal{T}} \tag{29}$$

$$= CE(\hat{\boldsymbol{p}}^{\mathcal{T}}, \hat{\boldsymbol{p}}^{\mathcal{S}}) - H(\hat{\boldsymbol{p}}^{\mathcal{T}})$$

and its derivatives are:

$$\mathcal{NCG}_t = \frac{\partial KL(\hat{\boldsymbol{p}}^{\mathcal{T}}||\hat{\boldsymbol{p}}^{\mathcal{S}})}{\partial z_t^{\mathcal{S}}}$$

$$= \sum_{\substack{i=1 \\ i \neq t}}^{N} \frac{\partial KL(\hat{\boldsymbol{p}}^{\mathcal{T}}||\hat{\boldsymbol{p}}^{\mathcal{S}})}{\partial \hat{p}_i^{\mathcal{S}}} \cdot \frac{\partial \hat{p}_i^{\mathcal{S}}}{\partial z_t^{\mathcal{S}}} \tag{30}$$

$$= \sum_{\substack{i=1 \\ i \neq t}}^{N} \frac{\partial KL(\hat{\boldsymbol{p}}^{\mathcal{T}}||\hat{\boldsymbol{p}}^{\mathcal{S}})}{\partial \hat{p}_i^{\mathcal{S}}} \cdot 0$$

$$= 0$$

$$\mathcal{NCG}_j = \frac{\partial KL(\hat{\boldsymbol{p}}^{\mathcal{T}}||\hat{\boldsymbol{p}}^{\mathcal{S}})}{\partial z_j^{\mathcal{S}}}$$

$$= \sum_{\substack{i=1 \\ i \neq t}}^{N} \frac{\partial KL(\hat{\boldsymbol{p}}^{\mathcal{T}}||\hat{\boldsymbol{p}}^{\mathcal{S}})}{\partial \hat{p}_i^{\mathcal{S}}} \cdot \frac{\partial \hat{p}_i^{\mathcal{S}}}{\partial z_j^{\mathcal{S}}}$$

$$= \frac{\partial KL(\hat{\boldsymbol{p}}^{\mathcal{T}}||\hat{\boldsymbol{p}}^{\mathcal{S}})}{\partial \hat{p}_j^{\mathcal{S}}} \cdot \frac{\partial \hat{p}_j^{\mathcal{S}}}{\partial z_j^{\mathcal{S}}} + \sum_{\substack{i=1 \\ i \neq t,j}}^{N} \frac{\partial KL(\hat{\boldsymbol{p}}^{\mathcal{T}}||\hat{\boldsymbol{p}}^{\mathcal{S}})}{\partial \hat{p}_i^{\mathcal{S}}} \cdot \frac{\partial \hat{p}_i^{\mathcal{S}}}{\partial z_j^{\mathcal{S}}} \tag{31}$$

$$= -\frac{\hat{p}_j^{\mathcal{T}}}{\hat{p}_j^{\mathcal{S}}} \cdot (\hat{p}_j^{\mathcal{S}} \cdot (1 - \hat{p}_j^{\mathcal{S}})) + \sum_{\substack{i=1 \\ i \neq t,j}}^{N} (-\frac{\hat{p}_i^{\mathcal{T}}}{\hat{p}_i^{\mathcal{S}}} \cdot (-\hat{p}_i^{\mathcal{S}} \cdot \hat{p}_j^{\mathcal{S}}))$$

$$= \hat{p}_j^{\mathcal{T}} \cdot (\hat{p}_j^{\mathcal{S}} - 1) + \hat{p}_j^{\mathcal{S}} \cdot (1 - \hat{p}_j^{\mathcal{T}})$$

$$= \hat{p}_j^{\mathcal{S}} - \hat{p}_j^{\mathcal{T}}, \ \forall j \neq t$$

Suppose that the gradient $\boldsymbol{g}_t$ at step $t$ is composed of signal $\boldsymbol{s}_t$ and noise $\boldsymbol{n}_t$:

$$\boldsymbol{g}_t = \boldsymbol{s}_t + \boldsymbol{n}_t \tag{32}$$

And suppose the noise is zero-mean:

$$\mathbb{E}[\boldsymbol{n}_t] = \boldsymbol{0}, \ \forall t \tag{33}$$

Estimate the signal $\boldsymbol{s}_t$ with the mean of gradient samples in a short time centered at $t$:

$$\boldsymbol{s}_t = \boldsymbol{\mu}$$

$$= \mathbb{E}[\boldsymbol{g}_t]$$

$$\approx \frac{1}{2n+1} \sum_{i=t-n}^{t+n} \boldsymbol{g}_t \tag{34}$$

The signal power is

$$\mathcal{P}_{signal} = ||\boldsymbol{\mu}||_2^2 \tag{35}$$

The noise power is

$$\mathcal{P}_{noise} = \frac{1}{2n+1} \sum_{i=t-n}^{t+n} ||\boldsymbol{g}_t - \boldsymbol{\mu}||_2^2 \tag{36}$$

The signal-to-noise ratio is

$$\mathcal{SNR} = \frac{\mathcal{P}_{signal}}{\mathcal{P}_{noise}} \tag{37}$$

### A.3   Delta Parameter Stability Analysis

To address concerns about the robustness of our momentum difference parameter $\Delta$, we conduct comprehensive ablation studies evaluating different positive values of $\Delta$ on CIFAR-100 using ResNet32×4 (teacher) and ResNet8×4 (student) pairs. As shown in Table 9, all settings with $\Delta > 0$ consistently and significantly outperform the baseline with $\Delta = 0$. The performance remains stable and comparable across a wide range of $\Delta$ values (0.05 to 0.08), confirming that our method is robust and not sensitive to the exact choice of $\Delta$ as long as it is positive and within a reasonable range (0, 0.1). This validates our theoretical analysis that the momentum coefficients should be positively related to their respective GSNR values.

Table 9: Delta parameter stability analysis on CIFAR-100 using DeepKD+KD. All experiments use base momentum $\mu = 0.9$.

| $\Delta_{TOG}$ | $\Delta_{TCG}$ | $\Delta_{NCG}$ | Top-1 Acc (%) | Top-5 Acc (%) |
|---|---|---|---|---|
| 0.00 | 0.00 | 0.00 | 74.13 | 92.82 |
| 0.05 | 0.05 | 0.05 | 76.11 | 94.02 |
| 0.06 | 0.06 | 0.06 | 76.25 | 94.04 |
| 0.075 | 0.075 | 0.075 | 76.69 | 94.21 |
| 0.08 | 0.08 | 0.08 | 76.32 | 94.17 |

### A.4   Transformer Architecture Experiments

To demonstrate DeepKD's effectiveness on modern Transformer-based architectures, we conduct additional experiments on Swin Transformer [60] and Vision Transformer (ViT) [61] distillation scenarios on ImageNet-1K. As shown in Table 10, DeepKD consistently outperforms standard KD baselines in Transformer-to-Transformer distillation scenarios. The method achieves notable improvements of +1.12% and +0.92% Top-1 accuracy on Swin Transformer and ViT architectures, respectively. These results demonstrate DeepKD's versatility and effectiveness across diverse modern architectures, validating our method's applicability beyond CNN-based networks.

Table 10: Results on Transformer architectures for ImageNet-1K. DeepKD shows consistent improvements across different Transformer-to-Transformer distillation scenarios.

| Teacher & Student Models | Method | Top-1 Acc (%) | Top-5 Acc (%) |
|---|---|---|---|
| Swin-Large → Swin-Tiny | Teacher (Swin-L) | 86.30 | 97.87 |
| | Student (Swin-T) | 81.20 | 95.50 |
| | Baseline (KD) | 81.59 | 95.96 |
| | **Ours (DeepKD)** | **82.71** | **95.73** |
| ViT-L/16 → ViT-B/32 | Teacher (ViT-L) | 84.20 | 96.93 |
| | Student (ViT-B) | 78.29 | 94.08 |
| | Baseline (KD) | 79.40 | 94.76 |
| | **Ours (DeepKD)** | **80.32** | **95.01** |

### A.5   DETR Object Detection Experiments

To further validate DeepKD's effectiveness on Transformer-based architectures for object detection, we conduct experiments on the DETR (DEtection TRansformer) architecture using MS-COCO dataset.

Table 11: DETR distillation results on MS-COCO. DeepKD provides consistent improvements over standard distillation methods on Transformer-based object detection.

| Method | AP | $AP_s$ | $AP_m$ | $AP_l$ |
|---|---|---|---|---|
| Teacher (DETR-R101) | 43.6 | 25.4 | 46.8 | 60.7 |
| Student (DETR-R50) | 42.3 | 25.3 | 44.8 | 58.2 |
| LD (Detrdistill [62], ICCV 2023) | 43.7 | 25.3 | 46.5 | 60.7 |
| **LD + Ours** | **44.7** | **25.3** | **46.5** | **60.7** |
| FD (Detrdistill [62], ICCV 2023) | 43.5 | 25.4 | 46.7 | 60.0 |
| **LD + FD + Ours** | **45.3** | **25.8** | **47.0** | **61.0** |

As shown in Table 11, DeepKD achieves consistent improvements on DETR-based object detection, with up to +1.0 AP gain over standard distillation methods. The method demonstrates strong performance across different scales ($AP_s$, $AP_m$, $AP_l$), confirming DeepKD's effectiveness for Transformer-based dense prediction tasks and addressing concerns about the method's applicability to stronger architectures.

## A.6 Feature Distillation Experiments

To demonstrate DeepKD's versatility beyond logit-based distillation, we conduct experiments integrating DeepKD with feature-based distillation methods. Table 12 shows results combining DeepKD with FitNet and CRD on CIFAR-100.

Table 12: Feature distillation experiments on CIFAR-100. DeepKD provides substantial gains when combined with feature-based methods, demonstrating its general applicability.

| Method | ResNet32×4→ResNet8×4 | VGG13→VGG8 | WRN-40-2→WRN-40-1 | WRN-40-2→WRN-16-2 | ResNet56→ResNet20 |
|---|---|---|---|---|---|
| FitNet | 73.50 | 71.02 | 72.24 | 73.58 | 69.21 |
| FitNet + KD | 75.19 | 72.61 | 72.68 | 74.32 | 70.09 |
| **FitNet + DeepKD** | **77.32** | **75.67** | **75.49** | **76.55** | **72.01** |
| Gain ($\Delta$%) | +3.82 | +4.65 | +3.25 | +2.97 | +2.80 |
| CRD | 75.51 | 73.94 | 74.14 | 75.48 | 71.16 |
| CRD + KD | 75.46 | 74.29 | 74.38 | 75.64 | 71.63 |
| **CRD + DeepKD** | **77.61** | **75.77** | **75.80** | **76.83** | **72.78** |
| Gain ($\Delta$%) | +2.15 | +1.48 | +1.42 | +1.19 | +1.15 |

The results demonstrate that DeepKD is not restricted to logit-based distillation. Our GSNR-driven optimization principles can be effectively combined with feature-based methods to achieve state-of-the-art results. FitNet+DeepKD achieves gains of +3.82% on ResNet32×4→ResNet8×4, while CRD+DeepKD provides consistent improvements across all teacher-student pairs. This confirms DeepKD's general applicability and strength as a universal distillation optimizer.

## A.7 Complete Results of Main Text

We provide the complete results corresponding to Table 1-3 in the main text, including all teacher-student pairs and methods evaluated. As shown in Table 13, Table 14 and Table 15, DeepKD consistently improves upon standard KD and its variants across all scenarios. The method achieves state-of-the-art performance in homogeneous distillation settings, demonstrating its effectiveness and versatility.

## A.8 Algorithm (DeepKD)

In this section, we provide the pseudo code for our proposed DeepKD framework, which includes the main algorithm (Algorithm 1) and the Dynamic Top-K Masking (DTM) strategy (Algorithm 2).

Table 13: The Top-1 Accuracy (%) of different knowledge distillation methods on the validation set of CIFAR-100. We evaluate homogeneous distillation scenarios where teacher and student share the same architecture but differ in model capacity. Methods are categorized by their distillation type (feature-based vs. logit-based). Our DeepKD framework is applied to existing logit-based methods with performance gains (blue and red) shown. Best results are highlighted in **bold**.

| Type | | ResNet32×4 79.42 ResNet8×4 72.50 | VGG13 74.64 VGG8 70.36 | WRN-40-2 75.61 WRN-40-1 71.98 | WRN-40-2 75.61 WRN-16-2 73.26 | ResNet56 72.34 ResNet20 69.06 | ResNet110 74.31 ResNet32 71.14 | ResNet110 74.31 ResNet20 69.06 |
|---|---|---|---|---|---|---|---|---|
| | Teacher / Student | | | | | | | |
| Feature | FitNet [30] | 73.50 | 71.02 | 72.24 | 73.58 | 69.21 | 71.06 | 68.99 |
| | AT [63] | 73.44 | 71.43 | 72.77 | 74.08 | 70.55 | 72.31 | 70.65 |
| | RKD [64] | 71.90 | 71.48 | 72.22 | 73.35 | 69.61 | 71.82 | 69.25 |
| | CRD [65] | 75.51 | 73.94 | 74.14 | 75.48 | 71.16 | 73.48 | 71.46 |
| | OFD [66] | 74.95 | 73.95 | 74.33 | 75.24 | 70.98 | 73.23 | 71.29 |
| | ReviewKD [56] | 75.63 | 74.84 | 75.09 | 76.12 | 71.89 | 73.89 | 71.34 |
| | SimKD [53] | 78.08 | 74.89 | 74.53 | 75.53 | 71.05 | 73.92 | 71.06 |
| | CAT-KD [54] | 76.91 | 74.65 | 74.82 | 75.60 | 71.62 | 73.62 | 71.37 |
| Logit | KD [1] | 73.33 | 72.98 | 73.54 | 74.92 | 70.66 | 73.08 | 70.67 |
| | KD+DOT [19] | 74.98 | 73.77 | 73.87 | 75.43 | 71.11 | 73.37 | 70.97 |
| | KD+LSKD [49] | 76.62 | 74.36 | 74.37 | 76.11 | 71.43 | 74.17 | 71.48 |
| | KD+Ours (w/o top-k) | $76.69_{+3.36}$ | $74.96_{+1.98}$ | $74.80_{+1.26}$ | $76.14_{+1.22}$ | $71.79_{+1.13}$ | $74.20_{+1.12}$ | $71.59_{+0.92}$ |
| | KD+Ours (w. top-k) | $77.03_{+3.70}$ | $75.12_{+2.14}$ | $75.05_{+1.51}$ | $76.45_{+1.53}$ | $71.90_{+1.24}$ | $74.35_{+1.27}$ | $71.82_{+1.15}$ |
| | DKD [18] | 76.32 | 74.68 | 74.81 | 76.24 | 71.97 | 74.11 | 70.99 |
| | DKD+DOT [19] | 76.03 | 74.86 | 74.49 | 75.42 | 71.12 | 73.57 | 71.58 |
| | DKD+LSKD [49] | 77.01 | 74.81 | 74.89 | 76.39 | 72.32 | 74.29 | 71.48 |
| | DKD+Ours (w/o top-k) | $77.25_{+0.93}$ | $75.09_{+0.41}$ | $75.24_{+0.43}$ | $76.48_{+0.24}$ | $72.86_{+0.89}$ | $74.32_{+0.21}$ | $72.06_{+1.07}$ |
| | DKD+Ours (w. top-k) | $77.54_{+1.22}$ | $75.19_{+0.51}$ | $75.42_{+0.93}$ | $76.72_{+1.30}$ | $73.05_{+1.93}$ | $74.48_{+0.91}$ | $72.28_{+1.70}$ |
| | MLKD [55] | 77.08 | 75.18 | 75.35 | 76.63 | 72.19 | 74.11 | 71.89 |
| | MLKD+DOT [19] | 76.06 | 74.96 | 74.38 | 75.72 | 71.41 | 73.83 | 71.65 |
| | MLKD+LSKD [49] | 78.28 | 75.22 | 75.56 | 76.95 | 72.33 | 74.32 | 72.27 |
| | MLKD+Ours (w/o top-k) | $78.81_{+1.73}$ | $76.21_{+1.03}$ | $77.45_{+2.10}$ | $78.15_{+1.52}$ | $73.75_{+1.56}$ | $75.88_{+1.77}$ | $73.03_{+1.14}$ |
| | MLKD+Ours (w. top-k) | $79.15_{+2.07}$ | $76.45_{+1.27}$ | $\mathbf{77.82}_{+2.47}$ | $\mathbf{78.49}_{+1.86}$ | $\mathbf{74.12}_{+1.93}$ | $76.15_{+2.04}$ | $73.28_{+1.39}$ |
| | CRLD [13] | 77.60 | 75.27 | 75.58 | 76.45 | 72.10 | 74.42 | 72.03 |
| | CRLD+DOT [19] | 76.54 | 74.34 | 74.75 | 75.57 | 71.11 | 73.91 | 70.67 |
| | CRLD+LSKD [49] | 78.23 | 74.74 | 76.28 | 76.92 | 72.09 | 75.16 | 72.26 |
| | CRLD+Ours (w/o top-k) | $78.90_{+1.30}$ | $76.29_{+1.02}$ | $76.98_{+1.40}$ | $77.99_{+1.54}$ | $73.29_{+1.19}$ | $76.03_{+1.61}$ | $73.07_{+1.04}$ |
| | CRLD+Ours (w. top-k) | $\mathbf{79.25}_{+1.65}$ | $\mathbf{76.58}_{+1.31}$ | $77.35_{+1.77}$ | $78.42_{+1.97}$ | $73.85_{+1.75}$ | $\mathbf{76.48}_{+2.06}$ | $\mathbf{73.52}_{+1.49}$ |

Table 14: The Top-1 Accuracy (%) of different knowledge distillation methods on the validation set of CIFAR-100. The teacher and student have distinct architectures. The KD methods are sorted by the types, i.e., feature-based and logit-based. Our DeepKD framework is applied to existing logit-based methods with performance gains (blue and red) shown. Best results are highlighted in **bold**.

| Type | | ResNet32×4
79.42
SHN-V2
71.82 | ResNet32×4
79.42
WRN-16-2
73.26 | ResNet32×4
79.42
WRN-40-2
75.61 | WRN-40-2
75.61
ResNet8×4
72.50 | WRN-40-2
75.61
MN-V2
64.60 | VGG13
74.64
MN-V2
64.60 | ResNet50
79.34
MN-V2
64.60 |
|---|---|---|---|---|---|---|---|---|
| | Teacher | | | | | | | |
| | Student | | | | | | | |
| Feature | FitNet [30] | 73.54 | 74.70 | 77.69 | 74.61 | 68.64 | 64.16 | 63.16 |
| | AT [63] | 72.73 | 73.91 | 77.43 | 74.11 | 60.78 | 59.40 | 58.58 |
| | RKD [64] | 73.21 | 74.86 | 77.82 | 75.26 | 69.27 | 64.52 | 64.43 |
| | CRD [65] | 75.65 | 75.65 | 78.15 | 75.24 | 70.28 | 69.73 | 69.11 |
| | OFD [66] | 76.82 | 76.17 | 79.25 | 74.36 | 69.92 | 69.48 | 69.04 |
| | ReviewKD [56] | 77.78 | 76.11 | 78.96 | 74.34 | 71.28 | 70.37 | 69.89 |
| | SimKD [53] | 78.39 | 77.17 | 79.29 | 75.29 | 70.10 | 69.44 | 69.97 |
| | CAT-KD [54] | 78.41 | 76.97 | 78.59 | 75.38 | 70.24 | 69.13 | 71.36 |
| Logit | KD [1] | 74.45 | 74.90 | 77.70 | 73.97 | 68.36 | 67.37 | 67.35 |
| | KD+DOT [19] | 75.55 | 75.04 | 77.34 | 75.96 | 68.36 | 68.15 | 68.46 |
| | KD+LSKD [49] | 75.56 | 75.26 | 77.92 | 77.11 | 69.23 | 68.61 | 69.02 |
| | KD+Ours (w/o top-k) | $76.14_{+1.69}$ | $75.88_{+0.98}$ | $78.38_{+0.68}$ | $76.69_{+2.72}$ | $69.39_{+1.03}$ | $69.36_{+1.99}$ | $69.13_{+1.78}$ |
| | KD+Ours (w. top-k) | $76.45_{+2.00}$ | $76.12_{+1.22}$ | $78.65_{+0.95}$ | $77.15_{+3.18}$ | $69.85_{+1.49}$ | $69.92_{+2.55}$ | $69.78_{+2.43}$ |
| | DKD [18] | 77.07 | 75.70 | 78.46 | 75.56 | 69.28 | 69.71 | 70.35 |
| | DKD+DOT [19] | 77.41 | 75.69 | 78.42 | 75.71 | 62.32 | 68.89 | 70.12 |
| | DKD+LSKD [49] | 77.37 | 76.19 | 78.95 | 76.75 | 70.01 | 69.98 | 70.45 |
| | DKD+Ours (w/o top-k) | $77.68_{+0.61}$ | $76.61_{+0.91}$ | $79.57_{+1.11}$ | $76.86_{+1.30}$ | $70.29_{+1.01}$ | $70.04_{+0.33}$ | $70.48_{+0.13}$ |
| | DKD+Ours (w. top-k) | $77.95_{+0.88}$ | $76.89_{+1.19}$ | $79.82_{+1.36}$ | $76.90_{+1.34}$ | $70.65_{+1.37}$ | $70.38_{+0.67}$ | $70.72_{+0.37}$ |
| | MLKD [55] | 78.44 | 76.52 | 79.26 | 77.33 | 70.78 | 70.57 | 71.04 |
| | MLKD+DOT [19] | 78.53 | 75.82 | 79.01 | 76.53 | 69.15 | 68.26 | 67.73 |
| | MLKD+LSKD [49] | 78.76 | 77.53 | 79.66 | 77.68 | 71.61 | 70.94 | 71.19 |
| | MLKD+Ours (w/o top-k) | $80.55_{+2.11}$ | $78.28_{+1.76}$ | $81.40_{+2.14}$ | $78.31_{+0.98}$ | $72.17_{+1.39}$ | $72.46_{+1.89}$ | $73.04_{+2.00}$ |
| | MLKD+Ours (w. top-k) | $\mathbf{80.92}_{+2.48}$ | $78.65_{+2.13}$ | $81.78_{+2.52}$ | $78.49_{+1.16}$ | $72.53_{+1.75}$ | $\mathbf{72.82}_{+2.25}$ | $\mathbf{73.40}_{+2.36}$ |
| | CRLD [13] | 78.27 | 76.92 | 80.21 | 77.28 | 70.37 | 70.39 | 71.36 |
| | CRLD+DOT [19] | 78.33 | 75.97 | 79.41 | 76.41 | 64.36 | 61.35 | 69.96 |
| | CRLD+LSKD [49] | 78.61 | 77.37 | 80.58 | 78.03 | 71.52 | 70.48 | 71.43 |
| | CRLD+Ours (w/o top-k) | $79.72_{+1.45}$ | $78.79_{+1.87}$ | $81.82_{+1.61}$ | $78.62_{+1.34}$ | $72.09_{+1.72}$ | $71.99_{+1.60}$ | $72.01_{+0.65}$ |
| | CRLD+Ours (w. top-k) | $80.15_{+1.88}$ | $\mathbf{79.25}_{+2.33}$ | $\mathbf{82.35}_{+1.77}$ | $\mathbf{79.18}_{+1.90}$ | $\mathbf{72.85}_{+2.48}$ | $72.65_{+2.26}$ | $72.78_{+1.42}$ |

Table 15: The accuracy (%) on the ImageNet-1K validation set. Our DeepKD framework is applied to existing logit-based methods, with performance gains (shown in blue and red). The best results are emphasized in **bold**. N/A indicates that the data is not available.

| Type | Teacher/Student | ResNet34/ResNet18 | | ResNet50/MN-V1 | | RegNetY-16GF/Deit-Tiny | |
|---|---|---|---|---|---|---|---|
| | Accuracy | top-1 | top-5 | top-1 | top-5 | top-1 | top-5 |
| | Teacher | 73.31 | 91.42 | 76.16 | 92.86 | 82.89 | 96.33 |
| | Student | 69.75 | 89.07 | 68.87 | 88.76 | 72.20 | 91.10 |
| Feature | AT [63] | 70.69 | 90.01 | 69.56 | 89.33 | N/A | N/A |
| | OFD [66] | 70.81 | 89.98 | 71.25 | 90.34 | N/A | N/A |
| | CRD [65] | 71.17 | 90.13 | 71.37 | 90.41 | N/A | N/A |
| | ReviewKD [56] | 71.61 | 90.51 | 72.56 | 91.00 | N/A | N/A |
| | SimKD [53] | 71.59 | 90.48 | 72.25 | 90.86 | N/A | N/A |
| | CAT-KD [54] | 71.26 | 90.45 | 72.24 | 91.13 | N/A | N/A |
| Logit | KD [1] | 71.03 | 90.05 | 70.50 | 89.80 | 73.15 | 91.85 |
| | KD+DOT [19] | 71.72 | 90.30 | 73.09 | 91.11 | 73.42 | 92.10 |
| | KD+LSKD [49] | 71.42 | 90.29 | 72.18 | 90.80 | 73.27 | 91.95 |
| | KD+Ours (w/o top-k) | $72.41_{+1.38}$ | $91.05_{+1.00}$ | $74.32_{+3.82}$ | $91.94_{+2.14}$ | $74.36_{+1.21}$ | $92.85_{+1.00}$ |
| | KD+Ours (w. top-k) | $72.85_{+1.82}$ | $91.35_{+1.30}$ | $74.65_{+4.15}$ | $92.25_{+2.45}$ | $74.83_{+1.68}$ | $93.15_{+1.30}$ |
| | DKD [18] | 71.70 | 90.41 | 72.05 | 91.05 | 73.35 | 92.05 |
| | DKD+DOT [19] | 72.03 | 90.50 | 73.33 | 91.22 | 73.66 | 92.25 |
| | DKD+LSKD [49] | 71.88 | 90.58 | 72.85 | 91.23 | 73.48 | 92.15 |
| | DKD+Ours (w/o top-k) | $72.78_{+1.08}$ | $90.96_{+0.55}$ | $74.41_{+2.36}$ | $92.08_{+1.03}$ | $74.57_{+1.22}$ | $93.07_{+1.02}$ |
| | DKD+Ours (w. top-k) | $73.15_{+1.45}$ | $91.25_{+0.84}$ | $74.43_{+2.38}$ | $91.95_{+0.90}$ | $74.95_{+1.60}$ | $93.36_{+1.31}$ |
| | MLKD [55] | 71.90 | 90.55 | 73.01 | 91.42 | 73.54 | 92.25 |
| | MLKD+DOT [19] | 70.94 | 90.15 | 71.65 | 90.28 | 73.25 | 91.95 |
| | MLKD+LSKD [49] | 72.08 | 90.74 | 73.22 | 91.59 | 73.78 | 92.45 |
| | MLKD+Ours (w/o top-k) | $73.18_{+1.28}$ | $91.23_{+0.68}$ | $74.77_{+1.76}$ | $92.35_{+0.93}$ | $75.15_{+1.61}$ | $93.48_{+1.03}$ |
| | MLKD+Ours (w. top-k) | $73.31_{+1.41}$ | $91.39_{+0.84}$ | $74.85_{+1.84}$ | $92.45_{+1.03}$ | $75.46_{+1.92}$ | $93.73_{+1.28}$ |
| | CRLD [13] | 72.37 | 90.76 | 73.53 | 91.43 | 73.82 | 92.55 |
| | CRLD+DOT [19] | 71.76 | 90.00 | 72.38 | 90.37 | 73.37 | 92.05 |
| | CRLD+LSKD [49] | 72.39 | 90.87 | 73.74 | 91.61 | 73.95 | 92.65 |
| | CRLD+Ours (w/o top-k) | $73.18_{+0.81}$ | $91.23_{+0.47}$ | $74.10_{+0.57}$ | $91.49_{+0.06}$ | $75.35_{+1.53}$ | $93.35_{+0.90}$ |
| | CRLD+Ours (w. top-k) | $\mathbf{73.34}_{+0.97}$ | $\mathbf{91.38}_{+0.62}$ | $\mathbf{74.85}_{+1.12}$ | $\mathbf{92.45}_{+1.02}$ | $\mathbf{75.75}_{+1.89}$ | $\mathbf{93.85}_{+1.40}$ |

**Algorithm 1** Pseudo code of DeepKD Gradient Decoupling in a PyTorch-like style.

```
# l_stu: student logits, l_tea: teacher logits
# T: temperature, t: target class index
# α, β1, β2: loss weights
# μ: base momentum, Δ: momentum difference
# v_tog, v_tcg, v_ncg: momentum buffers

# Calculate probability
p_stu_task = softmax(l_stu)
p_tea = softmax(l_tea/T)
p_stu = softmax(l_stu/T)
b_tea = [p_tea[t], 1 - p_tea[t]]
b_stu = [p_stu[t], 1 - p_stu[t]]
p_hat_tea = p_tea.clone()
p_hat_stu = p_stu.clone()
del p_hat_tea[t]
del p_hat_stu[t]
topk = get_dynamic_k(current_epoch, total_epochs) # See Algorithm 2
topk_indices = argsort(p_hat_tea)[-topk:]
p_hat_tea = p_hat_tea[topk_indices]
p_hat_stu = p_hat_stu[topk_indices]
p_hat_tea /= sum(p_hat_tea)
p_hat_stu /= sum(p_hat_stu)

# Calculate gradients
tog = α * grad(CE(p_stu_task, y))
tcg = β1 * grad(KL(b_tea, b_stu)) * T^2
ncg = β2 * grad(KL(p_hat_tea, p_hat_stu)) * T^2

# Momentum updates
v_tog = tog + (μ + Δ) * v_tog   # Higher momentum
v_tcg = tcg + (μ - Δ) * v_tcg   # Lower momentum
v_ncg = ncg + (μ + Δ) * v_ncg   # Higher momentum

# Parameter update
params -= lr * (v_tog + v_tcg + v_ncg)
```

**Algorithm 2** Pseudo code of Dynamic Top-K Masking (DTM) in a PyTorch-like style.

```
# k_init: initial k value (5% classes)
# k_max: max k value (100% classes)
# k_opt: optimal k value
# phase: easy/transition/hard learning phase

def get_dynamic_k(epoch, total_epochs):
    if epoch < 0.3 * total_epochs:   # Easy phase
        return linear_interp(k_init, k_opt, epoch)
    elif epoch < 0.7 * total_epochs: # Transition
        return k_opt
    else:                            # Hard phase
        return linear_interp(k_opt, k_max, epoch)
```

