# OpenReview forum: "DeepKD: A Deeply Decoupled and Denoised Knowledge Distillation Trainer"
_NeurIPS.cc/2025/Conference — NeurIPS 2025 poster_

### Official Review · Reviewer_Untf · 2025-07-01

**Clarity:** 3
**Significance:** 2
**Originality:** 2
**Rating:** 4
**Confidence:** 3

**Summary:**

This paper introduces DeepKD, a method that consists of two core components: a dual-level gradient decoupling mechanism based on GSNR analysis and dynamic top-k masking. The authors validate the effectiveness of DeepKD through experiments on CIFAR-100, ImageNet-1K, and MS-COCO, demonstrating performance improvements over baseline approaches.

**Questions:**

Provide justification for the chosen hyperparameters, such as using 200 iterations for GSNR computation - Include experiments with stronger teacher models, such as ViT or DETR-based architectures

**Ethical Concerns:**

["NO or VERY MINOR ethics concerns only"]

**Final Justification:**

Thank you to the authors for providing further explanations and conducting additional experiments that effectively addressed my main concerns. Based on these clarifications and new results, I have decided to raise my score. I hope that these insights and the outcomes of the new experiments will be incorporated into the final version of the paper, should it be accepted.

**Limitations:**

Yes

**Paper Formatting Concerns:**

generally okay.

**Quality:**

2

**Strengths And Weaknesses:**

Strengths:

1.The paper presents several straightforward yet effective ideas
2. Evaluation on the object detection task is a valuable addition

Weakness:

1. The proposed method introduces additional hyperparameters that are difficult to apply in practice, as some lack theoretical justification. For instance, computing GSNR every 200 iterations is hardcoded based on empirical observations. However, this choice likely depends on factors such as the dataset, model architecture, batch size, learning rate, and optimizer, which the authors do not thoroughly analyze.
2. Why are all experiments conducted using relatively weak teacher models, such as ResNet or R-CNN? How does the proposed method perform when applied to stronger architectures like ViT or DETR?

---

> ### Author Rebuttal · Authors · 2025-07-30
>
> **W1: The method introduces extra hyperparameters, such as the 200-iteration interval for GSNR computation, which seems empirically chosen and may depend on various factors. The authors do not fully analyze or justify these choices.**
>
> Thank you for raising this important point regarding hyperparameter justification. We would like to clarify that the GSNR computation interval (e.g., every 200 iterations) is not a hyperparameter involved in the actual training process, but rather a setting used exclusively for our offline theoretical analysis. The GSNR plots in our paper (e.g., Figure 1a, Figure 2) are generated to empirically validate our hypothesis about gradient signal quality, and the interval simply determines the sampling frequency for these visualizations. This is a standard approach for analyzing training dynamics and does not impact the method’s practical application.
>
> Our findings show that the theoretical conclusions drawn from the GSNR analysis are robust to the choice of sampling interval, as long as the interval is not extremely small (e.g., >50 iterations). A smaller interval may introduce more noise, while a larger interval improves statistical reliability at the cost of higher computational overhead. We chose 200 as a balanced value for clear visualization and efficient computation. Importantly, this choice does not affect the training or require tuning for different datasets, architectures, or optimization settings.
>
> In summary, our method does not introduce any GSNR-related hyperparameter that needs to be tuned in practice. The 200-iteration interval is solely for offline analysis and does not influence the training process. The actual hyperparameters used in our method are principled, practical, and robust, as demonstrated in our experiments. We will further clarify this distinction in the revised manuscript.
>
> ---
>
> **W2: Why not use stronger teacher models (e.g., ViT, DETR) in experiments? How does your method perform on these architectures?**
>
> Thank you for your question regarding the diversity and strength of teacher models in our experiments. Our experimental design follows established practice in the knowledge distillation literature, using widely adopted baselines such as ResNet and Faster R-CNN to ensure fair and direct comparison with prior state-of-the-art methods. In addition, we have included strong teacher models such as ResNet50 and RegNetY-16GF on large-scale datasets like ImageNet.
>
> We fully agree that evaluating on modern Transformer-based architectures is essential. In fact, our original submission already includes results on Vision Transformers: Table 3 demonstrates that DeepKD, when combined with MLKD, significantly improves the performance of a DeiT-Tiny student on ImageNet, with up to +1.90% Top-1 accuracy gain. We also evaluated strong teacher models, including RegNetY-16GF, across a range of modern architectures.
>
> To further strengthen our validation, we have conducted additional experiments on Swin Transformer (ICCV 2021) and ViT-to-ViT (ICLR 2021) distillation. As shown in Table 1 below, DeepKD achieves notable Top-1 accuracy improvements of +1.12% and +0.92%, respectively, on ImageNet-1K.
>
> _Table 1: Results on Larger Transformer Architectures (ImageNet-1K)_
>
> | Teacher & Student Models      | Method            | Top1 Acc (%) | Top5 Acc (%) |
> | :----------------------------| :---------------- | :-----------:| :-----------:|
> | Swin-Large → Swin-Tiny       | Teacher (Swin-L)  |     86.30    |     97.87    |
> |                              | Student (Swin-T)  |     81.20    |     95.50    |
> |                              | Baseline (KD)     |     81.59    |     95.96    |
> |                              | **Ours (DeepKD)** |   **82.71**  |   **96.23**  |
> | ViT-L/16 → ViT-B/32          | Teacher (ViT-L)   |     84.20    |     96.93    |
> |                              | Student (ViT-B)   |     78.29    |     94.08    |
> |                              | Baseline (KD)     |     79.40    |     94.76    |
> |                              | **Ours (DeepKD)** |   **80.32**  |   **95.01**  |
>
> In response to your specific question about DETR, we have also performed new experiments on the DETR architecture for object detection on MS-COCO. As shown in Table 2, DeepKD provides consistent improvements over standard distillation methods, achieving up to +1.0 AP gain on DETR-based models.
>
> _Table 2: DETR Distillation Results on MS-COCO_
>
> | Method                       |    AP    |   APs    |   APm    |   APl    |
> | :--------------------------- | :------: | :------: | :------: | :------: |
> | Teacher (DETR-R101)          |   43.6   |   25.4   |   46.8   |   60.7   |
> | Student (DETR-R50)           |   42.3   |   25.3   |   44.8   |   58.2   |
> | LD (Detrdistill, ICCV 2023)  |   43.7   |   25.3   |   46.5   |   60.7   |
> | **LD + Ours**                | **44.7** | **25.3** | **46.5** | **60.7** |
> | FD (Detrdistill, ICCV 2023)  |   43.5   |   25.4   |   46.7   |   60.0   |
> | **FD + Ours**           | **45.3** | **25.8** | **47.0** | **61.0** |
>
> These results demonstrate that DeepKD generalizes well to both CNN-based and Transformer-based architectures, including challenging tasks such as object detection with DETR.
>
> We believe these comprehensive experiments provide strong evidence of DeepKD’s scalability and effectiveness across a wide range of modern architectures. All new results will be included in our revised manuscript.
>
> ---
>
> **Q1: Why was 200 chosen for GSNR computation, and are there results on stronger teacher models like ViT or DETR?**
>
> Thank you for these questions. Please find a concise summary below, with detailed explanations provided in the main rebuttal text above:
>
> - **On Hyperparameter Justification:** As discussed in the section **"On Hyperparameters and Theoretical Justification,"** the GSNR computation interval is used only for offline analysis and does not affect training. Our method’s actual hyperparameters are principled, practical, and robust.
>
> - **On Experiments with Stronger Models:** As detailed in **"On the Strength and Diversity of Teacher/Student Architectures,"** we have conducted extensive experiments on modern and strong architectures, including DeiT, Swin Transformer, ViT, and DETR.
>
> We hope these clarifications fully address your concerns, and we will incorporate all new experimental results into our revised draft.

---

### Official Review · Reviewer_jZPz · 2025-07-03

**Clarity:** 3
**Significance:** 2
**Originality:** 3
**Rating:** 5
**Confidence:** 4

**Summary:**

This paper introduces DeepKD, a novel knowledge distillation method that combines principles from DOT and DKD. The core contribution is twofold.
1. It builds upon DOT's use of separate momentum hyperparameters for the cross-entropy and KL-divergence losses by introducing a principled approach to determine the momentum. This innovation removes the need to tune one of DOT's key hyperparameters. It extends this to the decoupled loss of DKD (DKD decouples of target and non-target knowledge).
2. The paper proposes Dynamic Top-K Masking (DTM), a technique designed to filter out noise from the probability distributions of semantically unrelated classes by masking out the low probability classes dynamically, in a scheduled manner.

**Questions:**

Question:
- What are the phase-1 and phase-2 values for ImageNet?


Minor Comment:
- It would be good to include some of the newer works like [A-E] in the appendix for a more comprehensive comparison. [A] also has a nice list of prior works to contextualize improvement over time.

[A] Yulei Niu, Long Chen, Chang Zhou, and Hanwang Zhang. "Respecting transfer gap in knowledge distillation." Advances in Neural Information Processing Systems, 2022.

[B] Tao Huang, Yuan Zhang, Mingkai Zheng, Shan You, Fei Wang, Chen Qian, and Chang Xu, “Knowledge diffusion for distillation,” Advances in Neural Information Processing Systems, 2023.

[C] Zheng Li, Xiang Li, Lingfeng Yang, Borui Zhao, Renjie Song, Lei Luo, Jun Li, and Jian Yang, “Curriculum temperature for knowledge distillation,” in Proceedings of the AAAI Conference on Artificial Intelligence, 2023, vol. 37, pp. 1504–1512.

[D] Yashas Malur Saidutta, Rakshith Sharma Srinivasa, Jaejin Cho, Ching-Hua Lee, Chouchang Yang, Yilin Shen, and Hongxia Jin, "CIFD: Controlled Information Flow to Enhance Knowledge Distillation." Advances in Neural Information Processing Systems, 2024

[E] Tao Huang, Shan You, Fei Wang, Chen Qian, and Chang Xu, “DIST+: Knowledge Distillation From a Stronger Adaptive Teacher,” IEEE TRANSACTIONS ON PATTERN ANALYSIS AND MACHINE INTELLIGENCE, VOL. 47, NO. 7, July 2025

**Ethical Concerns:**

["NO or VERY MINOR ethics concerns only"]

**Final Justification:**

Update after author discussion. My main concern was the instability of some of the results which the authors addressed by providing more experiments and rectifying the typo of results in the original submission.

**Limitations:**

Yes

**Quality:**

2

**Strengths And Weaknesses:**

**Strengths**
1. The paper explores a novel technique of dynamic top-K masking (DTM)
2. It explores a novel combination of DOT and DKD and proposes an automatic momentum hyper-param selection using GSNR.
3. The paper shows compelling empirical results, promising performance gains across a variety of student-teacher architectures and on several benchmark datasets, including CIFAR-100, ImageNet, and MS-COCO.

**Weaknesses**
1. A significant concern is the number of new hyperparameters with respect to KD. In addition to the $\Delta$ parameter for momentum, the method requires tuning $\Beta_1$ and $\Beta_2$ values (for TCKD and DTM, respectively) and a two-stage schedule (Phase-1 and Phase-2) for DTM.
2. The sensitivity of these DTM scheduling parameters is concerning. The ablation studies show very high performance swings. For instance, with k=55 and Phase-2=170, a minor change in the Phase-1 end from 60 to 40 results in a performance drop of over 7% (to 69.98%). This is significantly worse than the 72.50% accuracy achieved by a standard student model trained without any knowledge distillation. This raises questions about the method's stability and practicality.

---

> ### Author Rebuttal · Authors · 2025-07-30
>
> **W1: The main concern is the introduction of new hyperparameters, including `Δ`, `β₁`, `β₂`, and the DTM schedule (Phase-1 and Phase-2).**
>
> Thank you for raising this important point regarding the hyperparameters introduced by our method. We address each of the parameters as follows:
>
> 1. **Loss Weights (`β₁`, `β₂`):**
>    These are **not tuned hyperparameters** in our framework. To ensure fair and consistent comparisons, we directly adopt the fixed coefficients for TCKD and NCKD as proposed in the original DKD (CVPR 2022) method, following established best practices.
>
> 2. **Momentum Difference (`Δ`):**
>    This parameter is **principled and robust**. Our GSNR analysis provides strong theoretical motivation to simply set `Δ > 0`. As shown in our ablation study (**Table 5, Left**), the final performance is not sensitive to the exact value of `Δ` as long as it is positive; all settings with `Δ > 0` significantly and consistently outperform the baseline with `Δ=0`.
>
> 3. **DTM Schedule (`k-value`, `Phase-1`, `Phase-2`):**
>    We acknowledge that the DTM schedule introduces new hyperparameters, which is expected for a new mechanism. However, our empirical results show that the DTM schedule, implemented as a three-phase curriculum with a heuristic linear growth strategy inspired by **curriculum learning**, is highly robust and practical. This is validated by extensive experiments on the large-scale ImageNet-1K dataset, where the DTM component consistently improves performance across all tested teacher-student pairs, confirming both its stability and practical value.
>
> In summary, the key hyperparameters in DeepKD are either fixed by standard practice, guided by strong theoretical principles and shown to be robust, or empirically robust on large-scale tasks. This ensures that our framework remains practical and easy to apply.
>
> ---
>
> **W2: The ablation studies suggest DTM schedule parameters may cause large performance fluctuations. For example, changing Phase-1 from 60 to 40 (with k=55, Phase-2=170) leads to a >7% drop, even below the student-only baseline. This raises concerns about the method's stability and practicality.**
>
> We sincerely appreciate your careful and rigorous review. Your observation prompted us to conduct a thorough audit of our experiment logs, during which we discovered a critical data compilation error in Table 5 of our manuscript.
>
> We apologize for this oversight. The error was due to a manual mistake in transcribing results for two experimental settings. To ensure transparency and reproducibility, we will release our complete code and final model checkpoints soon.
>
> The two erroneous data points that led to concerns about model stability have now been corrected (see Table 1).
>
> _Table 1: Correction of erroneous data._
>  k-value | Phase1 | Phase2 | Top-1 | Top-5
> ---|---|---|---|---
>  40 | 55 | 170 | ~~69.98%~~ | 91.62%
>  40 | 55 | 170 | **76.98%** | 91.62%
>  60 | 60 | 170 | ~~70.19%~~ | 92.36%
>  60 | 60 | 170 | **77.19%** | 92.36%
>
> With these corrections, all tested configurations in our ablation study now significantly outperform the student-only baseline. The results are now logically consistent and demonstrate that our DTM framework is highly robust across a wide range of parameterizations. The previously reported "high performance swings" and "instability" were artifacts of the data error and are now resolved. The corrected results provide clear and consistent support for our claims.
>
> Thank you again for your diligence, which has been instrumental in identifying and correcting this issue, thereby strengthening the accuracy and reliability of our work.
>
> ---
>
> **Q1: What are the phase-1 and phase-2 values for ImageNet?**
>
> Thank you for your question regarding the DTM schedule parameters for our ImageNet experiments. Due to the high computational cost of exhaustive hyperparameter search on ImageNet, we adopted a principled two-step approach:
>
> 1. **Principled Schedule Transfer from CIFAR-100:**
>    We first transferred insights from our extensive CIFAR-100 ablations, which indicated that a well-proportioned three-phase curriculum is effective. Accordingly, for the 100-epoch ImageNet training, we set the phase transitions at the **25% mark (epoch 25)** and the **75% mark (epoch 75)**.
>
> 2. **Experimental Verification on ImageNet:**
>    We then conducted a targeted ablation study on ImageNet, exploring variations around these initial values. The results are summarized below:
>
> _Table 2: ImageNet DTM Schedule Verification (ResNet34 teacher, ResNet-18 student; static top-k: k=100 yields best performance)._
>  k-value | Phase1 (epochs) | Phase2 (epochs) | Top-1 Acc (%) | Top-5 Acc (%)
> ---|---|---|---|---
>  100 | 15 | 75 | 72.44 | 91.07
>  **100** | **25** | **75** | **72.85** | **91.35**
>  100 | 35 | 75 | 72.71 | 91.26
>  100 | 25 | 65 | 72.52 | 91.12
>  100 | 25 | 85 | 72.65 | 91.18
>
> This ablation confirms that our initial, principled choice is indeed optimal. The best-performing schedule used in our main ImageNet experiments is: **`k-value=100`, `Phase1` ends at epoch 25, and `Phase2` ends at epoch 75**. Importantly, the results demonstrate the **high robustness** of our DTM schedule on ImageNet, as performance remains stable across all tested configurations. This confirms that our method is not sensitive to its hyperparameters on large-scale tasks.
>
> ---
>
> **Minor Comments: Including recent works [A-E] in the appendix would provide a more complete comparison and better context for our improvements.**
>
> Thank you for these valuable literature suggestions. We agree that including these recent works [A-E] will help contextualize our contributions within the rapidly evolving field of knowledge distillation.
>
> In the revised manuscript, we will:
>
> - Update the **Related Work** section to thoroughly discuss these papers.
> - Expand the comparison tables in the **appendix** to include reported results from these methods where possible, providing a more comprehensive view of the state-of-the-art and better highlighting the improvements of our work over time.
>
> We appreciate your suggestions and are confident these additions will further strengthen our paper.

---

> ### Comment · Reviewer_jZPz · 2025-08-01
>
> I thank the authors for their detailed clarification! Thank you for sharing the rectified results on the DTM schedule ablation studies. That makes the proposed idea more robust than thought before.
>
> > the final performance is not sensitive to the exact value of Δ as long as it is positive; all settings with Δ > 0 significantly and consistently outperform the baseline with Δ=0.
>
> I did not see any results that confirm "all settings with Δ > 0 significantly and consistently outperform the baseline". There is only one setting with all \Delta_{\*} > 0, the last row.
> Do you have other results that looked at other positive combinations and found those accuracy results to be similar? Something like all $\Delta_{\*}=0.05 and all \Delta_{\*}=0.1 also yielded similar performance as \Delta_{\*}=0.075 as shown in the last row? Note: I am using \Delta_{\*} to represent all of \Delta_{TOG}, \Delta_{TCG}, \Delta_{NCG}

---

> > ### Author Response · Authors · 2025-08-01
> >
> > Thanks for your prompt response. We’re glad to hear that you appreciate our idea.
> >
> > To directly address your concern, we have conducted additional ablation experiments to systematically evaluate the effect of different positive values of Δ on the final performance. The results are summarized in Table 3 below. All experiments were performed with a base momentum value v = 0.9. We note that when Δ = 0.1, the momentum coefficient reaches 1 throughout training, which leads to gradient explosion and non-convergence. Therefore, we focus on Δ values in the range (0, 0.1).
> >
> > _Table 3: Ablation results for different Δ values using DeepKD+KD (ResNet32×4 teacher / ResNet8×4 student) on CIFAR-100._
> >
> >  ∆TOG | ∆TCG | ∆NCG | Top-1 Acc (%) | Top-5 Acc (%)
> > ---|---|---|---|---
> >  0.00 | 0.00 | 0.00 | 74.13 | 92.82
> >  0.05 | 0.05 | 0.05 | 76.11 | 94.02
> >  0.06 | 0.06 | 0.06 | 76.25 | 94.04
> >  0.075 | 0.075 | 0.075 | 76.69 | 94.21
> >  0.08 | 0.08 | 0.08 | 76.32 | 94.17
> >
> > As shown, all settings with Δ > 0 consistently and significantly outperform the baseline (Δ = 0). The performance remains stable and comparable across a wide range of Δ values, confirming that our method is robust and not sensitive to the exact choice of Δ as long as it is positive and within a reasonable range.
> >
> > Thank you again for your constructive feedback, which helped us further validate and clarify the robustness of our approach.

---

> ### Comment · Reviewer_jZPz · 2025-08-04
>
> Thank you for sharing the additional experiments. It would be good to add in the paper that experiments across Delta values are stable and add this experimental table in the appendix of the paper. I thank the authors for engaging in the review period. I have raised my score to reflect my view after the our discussions.

---

> > ### Author Response · Authors · 2025-08-04
> >
> > Thank you for your kind response and for revising your score. We appreciate your suggestion and will include the Delta stability results and the table in the appendix in the revised version. Thanks again for your thoughtful feedback during the review process.

---

### Official Review · Reviewer_qHHw · 2025-07-03

**Clarity:** 3
**Significance:** 3
**Originality:** 3
**Rating:** 4
**Confidence:** 3

**Summary:**

The paper proposes DeepKD, a novel knowledge distillation framework that integrates dual-level decoupling with adaptive denoising. The key contributions are: 1. Dual-Level Decoupling**: Decomposes gradients into task-oriented, target-class, and non-target-class components. Momentum coefficients are adaptively allocated based on gradient signal-to-noise ratio (GSNR), enabling independent optimization of different knowledge types. 2. Dynamic Top-k Masking: Gradually incorporates more non-target classes during training, filtering low-confidence logits to purify dark knowledge transfer. 3. Empirical Validation: Extensive experiments on CIFAR-100, ImageNet, and MS-COCO demonstrate state-of-the-art performance, showing DeepKD's effectiveness across diverse models and tasks.
DeepKD enhances knowledge transfer efficiency and model generalization through its theoretically grounded approach.

**Questions:**

1. How does the dynamic top-k masking mechanism specifically determine the scheduling of the k value during different training phases, and what is the impact of this scheduling on the overall training efficiency and model performance?
2. In terms of technical details, how does the dual-level decoupling strategy dynamically adjust the momentum coefficients for different knowledge components during the training process to ensure effective knowledge transfer and model optimization?

**Ethical Concerns:**

["NO or VERY MINOR ethics concerns only"]

**Final Justification:**

Considering the effectiveness and novelty of the method proposed in this article, combined with the opinions of other reviewers, I give the final boarderline acceptance rating.

**Limitations:**

Yes, the paper does discuss its limitations. It notes that the current work focuses on logit-based distillation, but the SNR-driven momentum decoupling mechanism could be extended to feature distillation scenarios. Additionally, the dynamic top-k masking strategy shows promise for improving distillation performance and warrants further exploration for different model architectures and datasets.

**Paper Formatting Concerns:**

not found

**Quality:**

3

**Strengths And Weaknesses:**

strengths:
1. theoretical contributions: The paper proposes a novel dual-level decoupling strategy based on gradient signal-to-noise ratio (GSNR) and a dynamic top-k masking mechanism, both of which are theoretically well-founded and address limitations in existing knowledge distillation methods.
2. empirical validation: Extensive experiments on multiple benchmarks (CIFAR-100, ImageNet, MS-COCO) demonstrate the effectiveness of DeepKD. The method shows consistent improvements across various model architectures and scenarios, including image classification and object detection.
3. innovation: the combination of GSNR-driven momentum allocation and dynamic top-k masking is original and provides a new perspective on optimizing knowledge distillation processes.
4. versatility: deepKD can be seamlessly integrated with existing logit-based distillation approaches and consistently achieves state-of-the-art performance.

Weaknesses:
1. limited to Logit-Based Distillation: The current framework focuses on logit-based distillation. While the authors mention potential extensions to feature-based scenarios, the paper does not provide concrete results or methods for these cases.
2. complexity: the introduction of multiple components (TOG, TCG, NCG) and the dynamic top-k masking mechanism may increase the complexity of the implementation and require careful tuning of hyperparameters.
3. scalability concerns: Although the method shows strong performance on the datasets tested, its scalability to even larger models or more diverse tasks remains to be fully explored and validated.
4.potential for overfitting: The detailed decoupling and masking strategies might lead to overfitting if not properly regularized, though the paper does not extensively discuss this aspect.

---

> ### Author Rebuttal · Authors · 2025-07-30
>
> **W1: The framework mainly discusses logit-based distillation, and lacks concrete results or methods for feature-based scenarios.**
>
> Thank you for highlighting the scope of our framework. To directly address this concern and further demonstrate the flexibility of DeepKD, we have conducted additional experiments on feature-based distillation.
>
> Specifically, we integrated DeepKD with two widely adopted feature-based methods, FitNet (ICLR 2015) and CRD (ICLR 2020), as shown in Table 1. Our objective was to evaluate whether DeepKD’s GSNR-driven optimization could enhance a combined feature-and-logit distillation pipeline, beyond what is achievable with standard KD.
>
> _Table 1: Feature Distillation Experiments  on the CIFAR-100 validation_
>  Method | ResNet32×4→ResNet8×4 | VGG13→VGG8 | WRN-40-2→WRN-40-1 | WRN-40-2→WRN-16-2 | ResNet56→ResNet20
> ---|---|---|---|---|---
>  FitNet | 73.50 | 71.02 | 72.24 | 73.58 | 69.21
>  FitNet + KD | 75.19 | 72.61 | 72.68 | 74.32 | 70.09
>  FitNet + DeepKD | 77.32 (+3.82%) | 75.67 (+4.65%) | 75.49 (+3.25%) | 76.55 (+2.97%) | 72.01 (+2.80%)
>  CRD | 75.51 | 73.94 | 74.14 | 75.48 | 71.16
>  CRD + KD | 75.46 | 74.29 | 74.38 | 75.64 | 71.63
>  CRD + DeepKD | 77.61 (+2.10%) | 75.77 (+1.83%) | 75.80 (+1.66%) | 76.83 (+1.35%) | 72.78 (+1.62%)
>
> These results provide clear evidence that DeepKD is **not restricted to logit-based distillation**. Our GSNR-based optimization can be effectively combined with feature-based approaches, consistently yielding state-of-the-art results. This demonstrates the broad applicability and strength of DeepKD as a general-purpose distillation optimizer.
>
> ---
>
> **W2: The framework introduces several components and a dynamic masking mechanism, which may raise concerns about implementation complexity and hyperparameter tuning.**
>
> Thank you for raising concerns about implementation complexity and hyperparameter tuning. We address these aspects separately below:
>
> 1. **Loss Weights (`β₁`, `β₂`):**
>    These coefficients are **not tuned** in our framework. We follow the standard settings from DKD (CVPR 2022) to ensure fair and consistent comparisons, using their fixed values for TCKD and NCKD loss components.
>
> 2. **Momentum Difference (`Δ`):**
>    This parameter is **theoretically motivated and robust**. Our GSNR analysis justifies simply setting `Δ > 0`. As shown in our ablation (Table 5, Left), the final performance is not sensitive to the exact value of `Δ`—any positive value following the GSNR principle consistently outperforms the baseline (`Δ=0`).
>
> 3. **DTM Schedule (`k-value`, `Phase-1`, `Phase-2`):**
>    While the DTM schedule introduces new hyperparameters, our experiments on large-scale datasets (e.g., ImageNet-1K) show that the schedule is highly robust and practical. The DTM consistently improves performance across all tested teacher-student pairs, confirming its stability and utility.
>
> In summary, DeepKD’s key hyperparameters are either fixed by standard practice, guided by strong theoretical principles, or shown to be robust in large-scale settings. This ensures the framework remains practical and easy to adopt.
>
> ---
>
> **W3: Concerns about scalability and potential overfitting: The scalability of the method to larger models and more diverse tasks, as well as the risk of overfitting due to decoupling and masking strategies, require further validation.**
>
> We appreciate your comments on scalability. To provide a comprehensive validation, we have conducted new experiments covering dataset scale, model scale, task diversity, and different distillation paradigms.
>
> Beyond classification and detection on **ImageNet-1K** and **MS-COCO** (see Tables 3 and 4 in our main paper), we further evaluated DeepKD on the challenging object detection task using the Transformer-based **DETR** architecture (see Table 2 below).
>
> _Table 2: DETR Distillation Results on MS-COCO_
>  Method | AP | APs | APm | APl
> ---|---|---|---|---
>  Teacher (DETR-R101) | 43.6 | 25.4 | 46.8 | 60.7
>  Student (DETR-R50) | 42.3 | 25.3 | 44.8 | 58.2
>  LD (Detrdistill, ICCV 2023) | 43.7 | 25.3 | 46.5 | 60.7
>  **LD + Ours** | **44.7** | **25.3** | **46.5** | **60.7**
>  FD (Detrdistill, ICCV 2023) | 43.5 | 25.4 | 46.7 | 60.0
>  **FD + Ours** | **45.3** | **25.8** | **47.0** | **61.0**
>
> We also evaluated DeepKD with strong teachers (e.g., **RegNetY-16GF**) and a diverse set of modern Transformer architectures. For example, DeepKD achieves a +1.90% gain when combined with MLKD on DeiT. Further, our new experiments on Swin Transformer (ICCV 2021) and ViT-to-ViT (ICLR 2021) distillation show notable Top-1 accuracy improvements of **+1.12%** and **+0.92%**, respectively (see Table 3).
>
> _Table 3: Larger Transformer Experiments on ImageNet-1K_
>  Teacher & Student Models | Method | Top1 Acc (%) | Top5 Acc (%)
> ---|---|---|---
> | Swin-Large → Swin-Tiny | Teacher (Swin-L) | 86.30 | 97.87
> |  | Student (Swin-T) | 81.20 | 95.50
> |  | Baseline (KD) | 81.59 | 95.96
> |  | **Ours (DeepKD)** | **82.71** | **96.23**
> | ViT-L/16 → ViT-B/32 | Teacher (ViT-L) | 84.20 | 96.93
> |  | Student (ViT-B) | 78.29 | 94.08
> |  | Baseline (KD) | 79.40 | 94.76
> |  | **Ours (DeepKD)** | **80.32** | **95.01**
>
>
> Additionally, we validated DeepKD on semantic segmentation, comparing with KA (Tong He, CVPR 2019), as shown below:
>
> _Table 4: Pascal VOC 2012 Semantic Segmentation (val set, trained on trainaug)_
>  Method | mIOU (%)
> ---|---
>  Teacher: ResNet-50 | 76.21
>  Student: MobileNetV2 | 70.57
>  Student+KD | 71.32
>  Student+FitNet | 71.30
>  Student+KA | 72.50
>  Student+Ours | **73.41**
>
> In summary, our extensive experiments—spanning large datasets, diverse tasks, strong teachers, a variety of modern Transformer architectures (DeiT, Swin, ViT, DETR), and multiple distillation paradigms—provide robust evidence of DeepKD’s scalability and generality.
>
> ---
>
> **W4: Overfitting risk: Could the decoupling and masking strategies cause overfitting if not well regularized? The paper does not discuss this in detail.**
>
> Thank you for raising the important issue of overfitting. We believe that the core mechanisms of DeepKD—GSNR-driven decoupling and Dynamic Top-k Masking (DTM)—are inherently regularizing and actually help mitigate overfitting.
>
> 1. **GSNR-Driven Decoupling as Regularization:**
>    Our GSNR-based momentum allocation stabilizes training by assigning higher momentum to more reliable gradient signals (TOG and NCG). This leads to convergence at **flatter minima** in the loss landscape, as shown in our loss landscape analysis (Figure 1(b)). Flatter minima are well-known to correlate with better generalization and reduced overfitting (Hao Li, NIPS 2018).
>
> 2. **DTM as Curriculum-Based Regularization:**
>    DTM filters out low-confidence, potentially noisy non-target logits, especially early in training. This prevents the student from overfitting to unreliable aspects of the teacher’s dark knowledge. By gradually expanding the knowledge set, DTM provides a curriculum learning effect, further enhancing generalization.
>
> Together, these mechanisms ensure that the student learns robust, generalizable features, actively guarding against overfitting.
>
> ---
>
> **Q1: How is the k value scheduled in DTM, and how does this affect training efficiency and model performance?**
>
> Thank you for your question regarding the scheduling of the k value in DTM.
>
> **1. DTM Scheduling Strategy:**
> Our DTM schedule is inspired by **curriculum learning** and is implemented in three phases using a linear growth heuristic:
>
> - **Phase 1 (Easy Learning):** `K` increases linearly from a small value (e.g., 5% of total classes) to an optimal static value (`k-value`). This allows the student to focus on the most confident, semantically relevant non-target classes at the start.
> - **Phase 2 (Transition):** `K` is held constant at `k-value`, giving the model time to consolidate knowledge from this core set.
> - **Phase 3 (Hard Learning):** `K` increases linearly from `k-value` to the full class count, exposing the student to the complete distribution of dark knowledge for final refinement.
>
> Empirically, we find it effective to end Phase 1 at ~25% of total epochs (e.g., epoch 60/240) and Phase 2 at ~70-75% (e.g., epoch 170-180/240).
>
> **2. Impact of DTM Scheduling:**
> - **Training Efficiency:** The scheduling logic is a simple epoch-based `if/else` and adds negligible computational overhead.
> - **Model Performance:** The curriculum-based schedule consistently improves performance, as shown in all main paper tables. For example, in Table 1, DTM improves `DKD+Ours` from 77.25% to **77.54%**. The approach prevents the student from being overwhelmed by noisy logits early on, leading to more stable and effective learning.
>
> ---
>
> **Q2: How does the dual-level decoupling strategy dynamically set the momentum coefficients for different knowledge components during training?**
>
> Thank you for this technical question. To clarify, our dual-level decoupling strategy does **not** dynamically adjust momentum coefficients during training. Instead, our GSNR analysis provides a **one-time, principled basis** for a fixed momentum allocation.
>
> 1. **Theoretical Basis:**
>    As shown in Figure 2(a) of the main paper, throughout training, the gradients for TOG and NCG consistently have higher GSNR than TCG. This indicates that TOG and NCG provide more reliable learning signals.
>
> 2. **Practical Mechanism:**
>    Based on this observation, we assign a **fixed higher momentum (`μ + Δ`)** to TOG and NCG, and a **fixed lower momentum (`μ - Δ`)** to TCG, as described in Equation 7. The value of `Δ` is constant and, as shown in our ablation, the method is robust to its precise value.
>
> This design achieves the benefits of GSNR-informed optimization without the complexity of dynamic adjustment, making DeepKD both effective and practical.
>
> We thank the reviewer again for their thoughtful feedback. We will revise the manuscript to clarify these points and are confident that these explanations address the concerns raised.

---

> ### Comment · Reviewer_qHHw · 2025-08-06
> **DeepKD: A Deeply Decoupled and Denoised Knowledge Distillation Trainer**
>
> Thank you for the author's detailed response, which has addressed some of my concerns, so I will keep my final rating unchanged.

---

> > ### Author Response · Authors · 2025-08-06
> >
> > Thanks for your time and thoughtful feedback. We sincerely appreciate your recognition of our work and your engagement during the review process.

---

### Official Review · Reviewer_ryqe · 2025-07-05

**Clarity:** 3
**Significance:** 3
**Originality:** 2
**Rating:** 4
**Confidence:** 4

**Summary:**

This work focuses on improving logit-based knowledge distillation. The authors firstly observe two critical issues: the conflicts between target-class and non-target-class knowledge flow, and the noise introduced by low-confidence "dark knowledge" from non-target classes. To mitigate these challenges, the paper introduces DeepKD, which integrates a dual-level decoupling mechanism with an adaptive denoising strategy. In particular, the proposed DeepKD assigns three independent momentum updaters for different gradient components (task-oriented, target-class, and non-target-class) according to the theoretical analysis of Gradient Signal-to-Noise Ratio (GSNR). Moreover, in order to effectively reduce the noise of non-target classes, a Dynamic Top-K Mask (DTM) mechanism is introduced to progressively filter low-confidence logits from both the teacher and student models during training. Extensive experiments are conducted on CIFAR-100 and ImageNet-1K on image classification task, and MS-COCO on object detection task.

**Questions:**

Please see the weaknesses above.

**Ethical Concerns:**

["NO or VERY MINOR ethics concerns only"]

**Final Justification:**

Given that my main concerns resolved, I’ve adjusted my score to 4.

**Limitations:**

Yes.

**Quality:**

3

**Strengths And Weaknesses:**

**Strengths**

1. The paper is well-motivated. The proposed DeepKD is pioneering in establishing a direct connection between Gradient Signal-to-Noise Ratio (GSNR) and momentum allocation in the context of logit-based knowledge distillation.

2. Independent momentum updaters derived from the GSNR analysis is proposed to prevent mutual interference among the different knowledge flows during training process.

3. The Dynamic Top-K Mask (DTM) mechanism is introduced to adaptively filter low-confidence logits from both the teacher and student models by gradually increasing the value of 'K'.

**Weaknesses**

1. The computational complexity is increased. The proposed DeepKD framework involves three distinct momentum updaters and a dynamic top-K mask mechanism with three predefined learning phases (Easy, Transition, Hard), inherently adds significant complexity to the training pipeline. It may lead to longer training time and potentially higher memory consumption compared to other KD methods.

2. The experimental settings are limited to CNN-based networks. The paper does not extensively demonstrate DeepKD's effectiveness or performance characteristics when applied to other dominant architectures, such Transformer-based networks.

3. The independent effectiveness assessment of proposed DeepKD is missing. DeepKD is frequently applied in conjunction with other logit-based knowledge distillation methods (e.g., KD, DKD) in the reported tables. This makes it challenging to precisely isolate and quantify DeepKD independent performance. Further empirical evidence is needed to strengthen its contributions, specifically by demonstrating its standalone effectiveness and the performance gains achieved when incorporating feature distillation.

---

> ### Author Rebuttal · Authors · 2025-07-30
>
> **W1: DeepKD introduces additional computational complexity, which may increase training time and memory usage compared to other KD methods.**
>
> Thank you for raising this important and practical concern regarding computational complexity. We analyze DeepKD's overhead on CIFAR-100 and ImageNet-1K. Results are shown in the tables below.
>
> _Table 1: Overhead analysis. For CIFAR-100, we use ResNet32×4 as the teacher and ResNet8×4 as the student, training on a single 2080Ti GPU for 240 epochs (480 for MLKD). For ImageNet-1K, we use ResNet34 as the teacher and ResNet-18 as the student, training on two RTX4090 GPUs for 100 epochs._
>  Method | CIFAR-100 Memory (M) | CIFAR-100 Time (Hours) | ImageNet-1K Memory (M) | ImageNet-1K Time (Hours)
> ---|---|---|---|---
>  KD | 799 | 1.6 | 21344 | 20.0
>  KD + Ours | 805 (+0.75%) | 2.6 (+62%) | 21370 (+0.12%) | 29.4 (+47%)
>  DKD | 799 | 1.7 | 21350 | 20.1
>  DKD + Ours | 805 (+0.75%) | 2.7 (+60%) | 21370 (+0.12%) | 29.8 (+48%)
>  MLKD | 983 | 9.2 | 34910 | 34.2
>  MLKD + Ours | 987 (+0.41%) | 12.3 (+33%) | 34960 (+0.14%) | 53.1 (+55%)
>  CRLD | 981 | 2.9 | 34862 | 32.2
>  CRLD + Ours | 985 (+0.41%) | 4.5 (+52%) | 34909 (+0.13%) | 51.2 (+59%)
>
> As shown, the practical overhead introduced by DeepKD is well-controlled and represents a favorable trade-off for the performance improvements achieved.
>
> **Memory Consumption is Minimal:**
> Table 1 demonstrates that DeepKD incurs only a negligible increase in memory usage—less than 1% on CIFAR-100 and less than 0.2% (about 10–25MB per GPU) on ImageNet-1K. This minor increase is primarily due to the storage of student gradients during distillation, as the teacher model remains frozen. The efficient CUDA memory allocator further mitigates any additional memory footprint, confirming that DeepKD is suitable for large-scale training scenarios.
>
> **Training Time and Efficiency:**
> Although DeepKD introduces a moderate increase in per-epoch training time, Table 2 demonstrates that it achieves superior accuracy with substantially fewer epochs than baseline methods. This results in a more favorable overall time-accuracy trade-off.  All DeepKD results are from **independent full training runs** at the specified epochs for fair comparison.
>
> _Table 2: DeepKD achieves higher accuracy with fewer training epochs. For fair comparison, all baseline methods are trained for 240 epochs on CIFAR-100 (480 for MLKD) and 100 epochs on ImageNet-1K, using ResNet32×4/ResNet8×4 and ResNet34/ResNet-18 as teacher/student pairs, respectively._
>  Method | CIFAR-100 Epoch | CIFAR-100 Top1 (%) | ImageNet-1K Epoch | ImageNet-1K Top1 (%)
> ---|---|---|---|---
>  KD | 240 | 73.33 | 100 | 71.03
>  KD+DOT | 240 | 74.98 | 100 | 71.72
>  KD+LSKD | 240 | 76.62 | 100 | 71.42
>  **KD+Ours** | **160** | **76.83** | **65** | **71.76**
>  DKD | 240 | 76.32 | 100 | 71.70
>  DKD+DOT | 240 | 76.03 | 100 | 72.03
>  DKD+LSKD | 240 | 77.01 | 100 | 71.88
>  **DKD+Ours** | **160** | **77.31** | **65** | **72.16**
>  MLKD | 480 | 77.08 | 100 | 71.90
>  MLKD+DOT | 480 | 76.06 | 100 | 70.94
>  MLKD+LSKD | 480 | 78.28 | 100 | 72.08
>  **MLKD+Ours** | **320** | **79.04** | **65** | **72.52**
>  CRLD | 240 | 77.6 | 100 | 72.37
>  CRLD+DOT | 240 | 76.54 | 100 | 71.76
>  CRLD+LSKD | 240 | 78.23 | 100 | 72.39
>  **CRLD+Ours** | **160** | **78.87** | **65** | **72.98**
>
> Our framework enables faster convergence and higher final accuracy than baseline methods, even with fewer training epochs. For example:
> - On CIFAR-100, **KD + Ours** achieves 76.83% accuracy in just 160 epochs, outperforming the 240-epoch KD baseline.
> - On ImageNet-1K, **CRLD + Ours** reaches 72.98% accuracy in only 65 epochs, surpassing the 100-epoch CRLD baseline.
>
> **Baselines Cannot Match DeepKD Even with Extended Training:**
> Table 3 further demonstrates that simply increasing the training duration for baseline methods does not close the performance gap with DeepKD.
>
> _Table 3: Baseline performance after extended training. For fair comparison, all methods are trained for 480 epochs on CIFAR-100 (960 epochs for MLKD) and 200 epochs on ImageNet-1K, as reported in the main paper._
>  Method | CIFAR-100 Epoch | CIFAR-100 Top1 (%) | ImageNet-1K Epoch | ImageNet-1K Top1 (%)
> ---|---|---|---|---
>  KD | 480 | 73.00 | 200 | 72.29
>  KD+DOT | 480 | 74.72 | 200 | 72.25
>  KD+LSKD | 480 | 76.51 | 200 | 71.99
>  **KD+Ours** | **240** | **77.03** | **100** | **72.85**
>  DKD | 480 | 75.89 | 200 | 72.10
>  DKD+DOT | 480 | 76.28 | 200 | 72.41
>  DKD+LSKD | 480 | 76.63 | 200 | 72.36
>  **DKD+Ours** | **240** | **77.54** | **100** | **73.15**
>  MLKD | 960 | 77.64 | 200 | 72.76
>  MLKD+DOT | 960 | 76.21 | 200 | 71.47
>  MLKD+LSKD | 960 | 78.87 | 200 | 72.96
>  **MLKD+Ours** | **240** | **79.15** | **100** | **73.31**
>  CRLD | 480 | 77.99 | 200 | 72.98
>  CRLD+DOT | 480 | 76.34 | 200 | 71.83
>  CRLD+LSKD | 480 | 78.61 | 200 | 73.13
>  **CRLD+Ours** | **240** | **79.25** | **100** | **74.23**
>
> For instance, even after 480 epochs, the CRLD baseline on CIFAR-100 only reaches 77.99%, which is still below our CRLD+Ours result of 79.25% achieved in half the epochs. The same trend is observed on ImageNet-1K.
>
> **The Training Cost is Justified by Substantial Gains:**
> The primary goal of knowledge distillation is to maximize the student model’s inference performance. In practice, it is standard and reasonable to invest more resources in a one-time training process if it results in a superior model, as inference will be performed many times over the model’s lifecycle. Importantly, this additional training cost does not affect the final inference speed of the student model. In summary, DeepKD provides a highly efficient route to a better-performing model.
>
> ---
>
> **W2: The experiments mainly focus on CNNs. More evidence is needed to show DeepKD's effectiveness on Transformer-based architectures.**
>
> Thank you for highlighting the importance of evaluating DeepKD on diverse architectures. We appreciate this point and have taken steps to provide more comprehensive evidence of DeepKD’s effectiveness on modern Transformer-based networks.
>
> To clarify, our original submission already included experiments on Vision Transformers, where DeepKD improved the Top-1 accuracy of a DeiT-Tiny student by up to +1.90% (see Table 3 in the main paper).
>
> To further address your concern, we conducted two additional sets of experiments, focusing on pure-Transformer architectures—Swin Transformer (ICCV 2021) and Vision Transformer (ViT, ICLR 2021)—on the ImageNet-1K dataset.
>
> _Table 4: Transformer Experiments on ImageNet-1K_
>  Teacher & Student Models | Method | Top-1 Acc (%) | Top-5 Acc (%)
> ---|---|---|---
> | Swin-Large → Swin-Tiny | Teacher (Swin-L) | 86.30 | 97.87
> |  | Student (Swin-T) | 81.20 | 95.50
> |  | Baseline (KD) | 81.59 | 95.96
> |  | **Ours (DeepKD)** | **82.71** | **96.23**
> | ViT-L/16 → ViT-B/32 | Teacher (ViT-L) | 84.20 | 96.93
> |  | Student (ViT-B) | 78.29 | 94.08
> |  | Baseline (KD) | 79.40 | 94.76
> |  | **Ours (DeepKD)** | **80.32** | **95.01**
>
> As shown, DeepKD consistently and significantly outperforms the standard KD baseline in these challenging Transformer-to-Transformer distillation scenarios.
>
> ---
>
> **W3: DeepKD is mainly evaluated together with other logit-based distillation methods, making it hard to isolate its individual effect. More evidence is needed to show its independent contribution and its impact when combined with feature distillation.**
>
> Thank you for this insightful question regarding the independent assessment of DeepKD and its applicability beyond logit-based methods. We would like to clarify the nature of our framework and provide new experimental results to directly address your concerns.
>
> **First**, DeepKD is a training framework or optimization strategy, not a standalone loss function. This is a common characteristic among many state-of-the-art knowledge distillation methods, such as DKD (CVPR 2022), DOT (ICCV 2023), and LSKD (CVPR 2024). These methods are designed to enhance a primary distillation signal, either by reformulating the loss, modifying the optimization process, or pre-processing the logits. As such, none of these methods—including DeepKD—can be meaningfully evaluated in complete isolation from a base teacher-student setup.
>
> **Second**, the independent effectiveness of our two main contributions—Decoupled Gradients and Dynamic Top-k Masking (DTM)—is already quantified in Table 6 of our main paper, which reports the individual performance gains from each component across various logit-based baselines.
>
> **Finally**, to further demonstrate the versatility of DeepKD, we conducted new experiments by integrating DeepKD with two widely used feature-based distillation methods: FitNet (ICLR 2015) and CRD (ICLR 2020). The results, summarized in Table 5, show that DeepKD provides substantial additional gains over standard feature distillation and logit-based KD combinations.
>
> _Table 5: Feature Distillation Experiments on the CIFAR-100 validation._
>  Method | ResNet32×4→ResNet8×4 | VGG13→VGG8 | WRN-40-2→WRN-40-1 | WRN-40-2→WRN-16-2 | ResNet56→ResNet20
> ---|---|---|---|---|---
>  FitNet | 73.50 | 71.02 | 72.24 | 73.58 | 69.21
>  FitNet + KD | 75.19 | 72.61 | 72.68 | 74.32 | 70.09
>  FitNet + DeepKD | 77.32 (+3.82%) | 75.67 (+4.65%) | 75.49 (+3.25%) | 76.55 (+2.97%) | 72.01 (+2.80%)
>  CRD | 75.51 | 73.94 | 74.14 | 75.48 | 71.16
>  CRD + KD | 75.46 | 74.29 | 74.38 | 75.64 | 71.63
>  CRD + DeepKD | 77.61 (+2.10%) | 75.77 (+1.83%) | 75.80 (+1.66%) | 76.83 (+1.35%) | 72.78 (+1.62%)
>
> These results provide strong evidence that DeepKD is not limited to logit-based distillation. Its GSNR-driven optimization principles can be effectively combined with feature-based methods, demonstrating the generality and strength of our framework. We will include these findings in the appendix of our revised manuscript.
>
> We hope these clarifications and new results comprehensively address your concerns. We will revise our manuscript to make these points more explicit and appreciate your thoughtful feedback.

---

### Decision · Program_Chairs · 2025-09-17

**Decision:**

Accept (poster)

**Comment:**

This submission introduces DeepKD, a novel approach to logit-based knowledge distillation, which leverages a dual-level decoupling mechanism and an adaptive denoising strategy to mitigate the observed issues of target-class and non-target-class knowledge conflict and the noise from low-confidence logits. The method has been empirically validated across several benchmarks, including CIFAR-100, ImageNet-1K, and MS-COCO, showing promising results.

Taking into consideration the reviews provided, the submission is positioned on the borderline between accept and reject. While the novelty and theoretical contributions of DeepKD are compelling, and its empirical validation demonstrates effectiveness, concerns around its computational complexity, the limited scope of architectures evaluated, and the absence of an independent effectiveness assessment of DeepKD temper its overall impact.

However, given the positive adjustments in response to reviewer concerns, especially the additional experiments and clarifications provided, it seems that the paper has a solid technical foundation and contributes positively to the advancement of knowledge distillation methods. Therefore, the decision leans towards a "Accept". It is recommended that the authors incorporate the insights and outcomes from the additional experiments into the final version of the paper. Moreover, addressing the highlighted weaknesses, particularly around computational complexity and expanding the scope of experimental validation to include other architectures, would substantially strengthen the paper.